

# Mining software insights: uncovering the frequently occurring issues in low-rating software applications

Nek Dil Khan[1], Javed Ali Khan[2], Jianqiang Li[1], Tahir Ullah[3] and Qing Zhao[1]

[1] Faculty of Information Technology, Beijing University of Technology, Beijing, China
[2] Department of Computer Science, School of Physics, Engineering and Computer Science, University of Hertfordshire, Hatfield, United Kingdom
[3] Faculty of Computer Science and Technology, Beijing Institute of Technology, Haidian, Beijing, China

## ABSTRACT

In today's digital world, app stores have become an essential part of software distribution, providing customers with a wide range of applications and opportunities for software developers to showcase their work. This study elaborates on the importance of end-user feedback for software evolution. However, in the literature, more emphasis has been given to high-rating & popular software apps while ignoring comparatively low-rating apps. Therefore, the proposed approach focuses on end-user reviews collected from 64 low-rated apps representing 14 categories in the Amazon App Store. We critically analyze feedback from low-rating apps and developed a grounded theory to identify various concepts important for software evolution and improving its quality including user interface (UI) and user experience (UX), functionality and features, compatibility and device-specific, performance and stability, customer support and responsiveness and security and privacy issues. Then, using a grounded theory and content analysis approach, a novel research dataset is curated to evaluate the performance of baseline machine learning (ML), and state-of-the-art deep learning (DL) algorithms in automatically classifying end-user feedback into frequently occurring issues. Various natural language processing and feature engineering techniques are utilized for improving and optimizing the performance of ML and DL classifiers. Also, an experimental study comparing various ML and DL algorithms, including multinomial naive Bayes (MNB), logistic regression (LR), random forest (RF), multi-layer perception (MLP), k-nearest neighbors (KNN), AdaBoost, Voting, convolutional neural network (CNN), long short-term memory (LSTM), bidirectional long short term memory (BiLSTM), gated recurrent unit (GRU), bidirectional gated recurrent unit (BiGRU), and recurrent neural network (RNN) classifiers, achieved satisfactory results in classifying end-user feedback to commonly occurring issues. Whereas, MLP, RF, BiGRU, GRU, CNN, LSTM, and Classifiers achieved average accuracies of 94%, 94%, 92%, 91%, 90%, 89%, and 89%, respectively. We employed the SHAP approach to identify the critical features associated with each issue type to enhance the explainability of the classifiers. This research sheds light on areas needing improvement in low-rated apps and opens up new avenues for developers to improve software quality based on user feedback.

Corresponding author
Qing Zhao, zhaoqing@bjut.edu.cn

# INTRODUCTION

End-user feedback is a critical resource in software evolution, offering informative insights into user needs, frequently occurring issues, and preferences. The rise of platforms like Amazon, Google Play, iPhone app stores, user forums, and Twitter has become a significant source for gathering user comments and intertwined informative information related to software evolution (*Sarı, Tosun & Alptekin, 2019*; *Ali Khan et al., 2018*; *Maalej et al., 2016*; *Khan et al., 2019a*). This feedback helps comprehend user needs and preferences, thus aiding system designers and developers in enhancing software functionality (*Wang et al., 2019*). However, the daily influx of user reviews on social media platforms leads to an information overload challenge (*Mao et al., 2017*). Where feedback mechanisms on these platforms enable users to rate the usefulness of reviews (*Malik, 2020*) resulting in a stream of information. These analyses are crucial for understanding user requirements, app design, and debugging, and they play a significant role in the evolution of software (*Dąbrowski et al., 2022*; *Wouters et al., 2022*). Recent research has mainly focused on extracting valuable insights from end-user reviews related to new features (*Sorbo et al., 2015*; *Dhinakaran et al., 2018*), non-function requirements (*Bakiu & Guzman, 2017*), feature requests & end-user rationale (*Kurtanović & Maalej, 2018*; *Ullah et al., 2023*) and informed decision-making (*Ali Khan et al., 2020*; *Sarro et al., 2018*) by analyzing feedback from user reviews (*Sarro et al., 2018*), tweets (*Guzman, Ibrahim & Glinz, 2017*), and forum posts (*Khan, Liu & Wen, 2020*; *Morales-Ramirez, Kifetew & Perini, 2017*; *Khan et al., 2024b*). However, according to our knowledge, to date, less work has been reported to date on identifying software requirements (*Ebrahimi, Tushev & Mahmoud, 2021*; *Garousi & Mäntylä, 2016*; *Guzman, Ibrahim & Glinz, 2017*) software bugs or issues (*Maalej & Nabil, 2015*; *Stanik, Haering & Maalej, 2019*) for low-rating software applications (*Ullah et al., 2023*; *Sarro et al., 2018*).

Moreover, end-user feedback is crucial in software bug detection, including user experience (UX)-related issues, which are integral to software development and success. However, in software engineering literature (*Tabassum et al., 2023*; *Zhang et al., 2016*), emphasis has been given to bug detection by exploring software artifacts such as bug reports, source codes, and change history for large-scale software applications. Researchers have proven that bug identification and resolution are critical steps in enhancing software reliability and efficiency (*Leinonen & Roto, 2023*). The UX aspect, containing design and usability issues, is closely linked to software bugs, as unresolved bugs can lead to poor UX. A detailed literature study on automated debugging and bug-fixing solutions highlights the growing direction of incorporating UX considerations into the process, acknowledging that user satisfaction is as crucial as technical correctness in software development

(*Dąbrowski et al., 2022*). This paradigm shift, where user experience is provided equal importance to technical functionality, is necessary for creating user-centred Software, a concept further underlined by research on service design handover to UX design and the challenges of bug detection and prevention in enhancing overall software quality (*Leinonen & Roto, 2023*). However, these research studies are limited to a large software application that analyzes traditional software artifacts to identify and process software bugs. Recently, due to the emergence of social media, end-users frequently submit various software bugs and issues related to UX as feedback, which needs concentration from the software vendors to satisfy end-user needs. Therefore, by tapping into these rich sources of feedback, developers can gain a more comprehensive understanding of the strengths and weaknesses of their applications, allowing for more targeted and practical improvements. Thus, incorporating input from these various sources is becoming important in the iterative software development method, pushing innovation and improving user satisfaction (*Jeong & Lee, 2022*).

Recent studies have analyzed different strategies for automatic software issue reports, leveraging machine learning (ML) algorithms to improve the efficiency and accuracy of this process (*Afric et al., 2023*). This methodological shift is essential, as it allows for managing large volumes of data with increased precision, thereby enhancing the software evolution cycle (*Pandey et al., 2017*). Also, identifying issues is vital for technological remediation, as well as considering the overall impact on user experience and satisfaction (*Cho, Lee & Kang, 2022*). For instance, distinguishing between a critical security fault and a small usability bug is crucial for proper response procedures. By integrating a range of techniques, as discussed in the literature (*Pandey et al., 2017*; *Cho, Lee & Kang, 2022*), the goal is to create software developments that are more functional and focused on the user's experience. This approach ensures that the technical quality and the user's interaction with the software are equally important. This motivated us to focus on issue classification in software development which is considered a critical focus of the proposed study. Classifying software issues beyond essential detection requires carefully analyzing multiple factors such as severity, nature, and possible impact on the end user. This analytical approach is necessary to prioritize issues, allocate resources virtually, and drive evolution toward more powerful and user-friendly software solutions (*Cho, Lee & Kang, 2022*).

In contrast, aligning with the previous research findings (*Liang et al., 2012*), the proposed research analyses end-user feedback from the Amazon Software App (ASA) store for the low-rating application to explore commonly & frequently reported bugs and group them into different categories by utilising different ML and deep learning (DL) models. The proposed approach helps software vendors and developers identify critical information considered pivotal for enhanced user satisfaction and improved quality, such as performance, user interface (UI), UX, functionality, compatibility, customer support, and security issues. Moreover, we delve into the often-overlooked domain of low-rated applications in the ASA Store, studying 64 apps across 14 categories. The study aims to find the possible reasons behind their low ratings, which are essential for software

developers & vendors to improve app performance and user satisfaction. Mainly the key contributions of the proposed approach are:

- Curated a novel research dataset of end-user feedback from the ASA store representing frequently occurring issue types in the software apps
- Proposed novel grounded theory by critically analyzing end-user feedback to identify the frequently occurring issues type UI and UX, functionality and features, compatibility and device-specific, customer support and responsiveness, and security and privacy issues.
- Developed a truth set for software issues and their types using a content analysis approach
- Employing a series of fine-tuned baseline ML and DL classifiers such as multinomial naive Bayes (MNB), logistic regression (LR), random forest (RF), multi-layer perception (MLP), k-nearest neighbors (KNN), AdaBoost, Voting, convolutional neural network (CNN), long short-term memory (LSTM), bidirectional long short term memory (BiLSTM), gated recurrent unit (GRU), bidirectional gated recurrent unit (BiGRU), and recurrent neural network (RNN) to report their performance in identifying various issue types.
- To improve the explainability of ML classifiers (MLP), we employed the SHAP approach to identify the critical features associated with each issue type. It will help software vendors and developers understand the complex decision-making of ML classifiers.

### The rest of the article is arranged as follows

"Related Work" provides the relevant literature study; "Proposed Methodology" discusses the research methodology; the section concentrates on the annotation recension to identify issue types; "Automated Classification of End-User Feedback into Issue Types" elaborates on the automated classification of the crowd user comments into different issue types and shows the results. "Discussion" discusses the details of research findings and threats to validity. "Conclusion and Future Work" concludes the article and emphasizes the future direction.

## RELATED WORK

The analysis of end-user reviews for software evolution has become an increasingly vital area in software engineering, as these reviews often contain critical insights into real-world application performance and user experience. Recently, software researchers have proposed many research approaches involving the analysis of end-user feedback to extract information related to software evolution, such as new features, non-functional requirements, and issues. Moreover, *Zhang et al. (2016)*, *Liang et al. (2012)* and *Dąbrowski et al. (2022)* conducted a detailed systematic literature study (SLR) on bug detection by considering bug reports, source codes, and change history artefacts. In contrast, the core emphasis of the proposed approach is on exploring end-user reviews for issue-related

information and their possible sub-types. Below, we provide the state-of-the-art related work and then compare the proposed with them.

Dąbrowski et al. (2022) highlight the breadth of research in this area, emphasizing the importance of analyzing app reviews for insights into software evolution. It also provides insights into how these analyses can benefit researchers and commercial organizations in developing app review analysis techniques. Malgaonkar, Licorish & Savarimuthu (2022) study further underscores this by prioritizing user issues in app reviews, which often include requests for new features, bug fixes, and enhancements, providing a direct link to potential software issues questions and trends in this field and the impact of these studies (Garousi & Mäntylä, 2016). Additionally, the role of ML in software fault prediction, as analyzed in an SLR, indicates a growing trend towards automated and predictive approaches in identifying software bugs, aligning with the needs and expectations of software engineers and end-users (Khan et al., 2023b, 2017). Moreover, Hai et al. (2022) analysis of cloud-based bug-tracking software faults research using ML approaches highlights the valuable applications of these methodologies in minimizing the time and cost needed for software testing.

Furthermore, Iacob & Harrison (2013), Zhao et al. (2021) and Guzman & Maalej (2014) have highlighted the significance of studying user reviews to enhance software quality. They claim that end-user reviews, particularly in digital platforms like app stores, provide rapid insights into user needs and experiences, which is vital for iterative software modification. Further, works by Iacob & Harrison (2013) and Carreno & Winbladh (2013) highlight the utility of end-user feedback in identifying specific software issues like bugs, feature requests, and user experience issues. The present analysis employs a grounded theory and content analysis method to analyze user input, which aligns with the methods used by other studies. The importance of grounded theory in extracting meaningful patterns from qualitative data, such as user evaluations, is highlighted by Maalej & Robillard (2013) and Strauss & Corbin (1998).

Additionally, methods like those of Neuendorf (2017) in content analysis deliver a systematic framework for classifying and examining end-user reviews to extract actionable insights. The focus on low-rated software applications, as objected to the more often examined high-rated software applications, is a novel aspect of this analysis. Sarro et al. (2018) and Morales-Ramirez, Kifetew & Perini (2017) have primarily focused on high-rated applications, thus requiring an essential understanding of low-rated software feedback. This gap is handled in the current study by systematically examining low-rated applications to comprehend the underlying causes of user dissatisfaction. In utilising ML and DL models to analyse user feedback, Khan, Liu & Wen (2020), Ullah et al. (2023) and Kurtanović & Maalej (2017) are specifically suitable. They have applied various ML and DL techniques, including those used in this study, like MNB, LR, RF, and MLP, to classify and analyze user feedback for software improvement. The effectiveness of these algorithms in processing large volumes of user data is corroborated by Alkadhi et al. (2017), who applied similar techniques to extract meaningful insights from user comments.

Unlike the state-of-the-art methodologies mentioned above, the proposed approach analyzes the end-user feedback to extract issues-related information from the ASA store.

We argue that such information is critical for software evolution and end-user satisfaction. If not considered timely for software evolution, it results in uninstalling and leaving the software application. Moreover, the identified feedback issues are further classified into compatibility & device, functionality and feature, customer Support & responsiveness, security and privacy, UI and UX, and performance & stability to help software developers better understand the frequently occurring issues with the software applications. We believe the proposed approach is the first step in understanding the software issues extracted from end-user reviews for low-ranked software apps compared to the previous approaches that focus more on extracting new features and considering high-rating popular applications.

## PROPOSED METHODOLOGY

This section first discusses the proposed research questions that the proposed research approach attempted to solve. Secondly, we elaborated on the proposed research methodology and its constituent sub-steps to identify frequently occurring issues and their categories.

### Proposed research questions

In this article, we aim to explore end-users comments in the ASA store by focusing on low-ranked software applications and capturing frequently occurring issues and their subtypes. This can help software developers to possibly improve the performance of low-ranked software applications by timely incorporating the useful information in the software evolution process and employing different ML and DL algorithms to automatically capture and identify issues for the software developer and requirements engineers to improve the existing functionalities of low-ranked software applications. Below, we formulated research questions that we are interested in answering using the proposed research methodology.

**RQ1:** How do end-users express their grudges in the ASA store about low-rated software applications?

**RQ2:** What different types of frequently occurring issues can be identified in the ASA store?

**RQ3:** How effectively can ML and DL algorithms analyze and categorize end-user issues or bugs for low-rated apps?

In RQ1, we perform a thorough and critical manual analysis of end-user comments collected from the ASA store to identify how user submit their opinions on frequently occurring issues for the software applications under discussion. This research culminates in the development of a novel grounded theory for identifying issues and their types. For RQ2, using the developed grounded theory and content analysis approach, we foresee identifying prevalent issue types by analyzing end-user feedback reported against the low-rated software applications. This results in the development of a ground truth, which is mandatory for training and validating ML and DL classifiers. Moving on to RQ3, the proposed approach focuses on evaluating and comparing the performance of various ML and DL classifiers on automatically detecting and classifying the end-user comments to various issue types aiming at optimizing the efficiency and scalability of the proposed

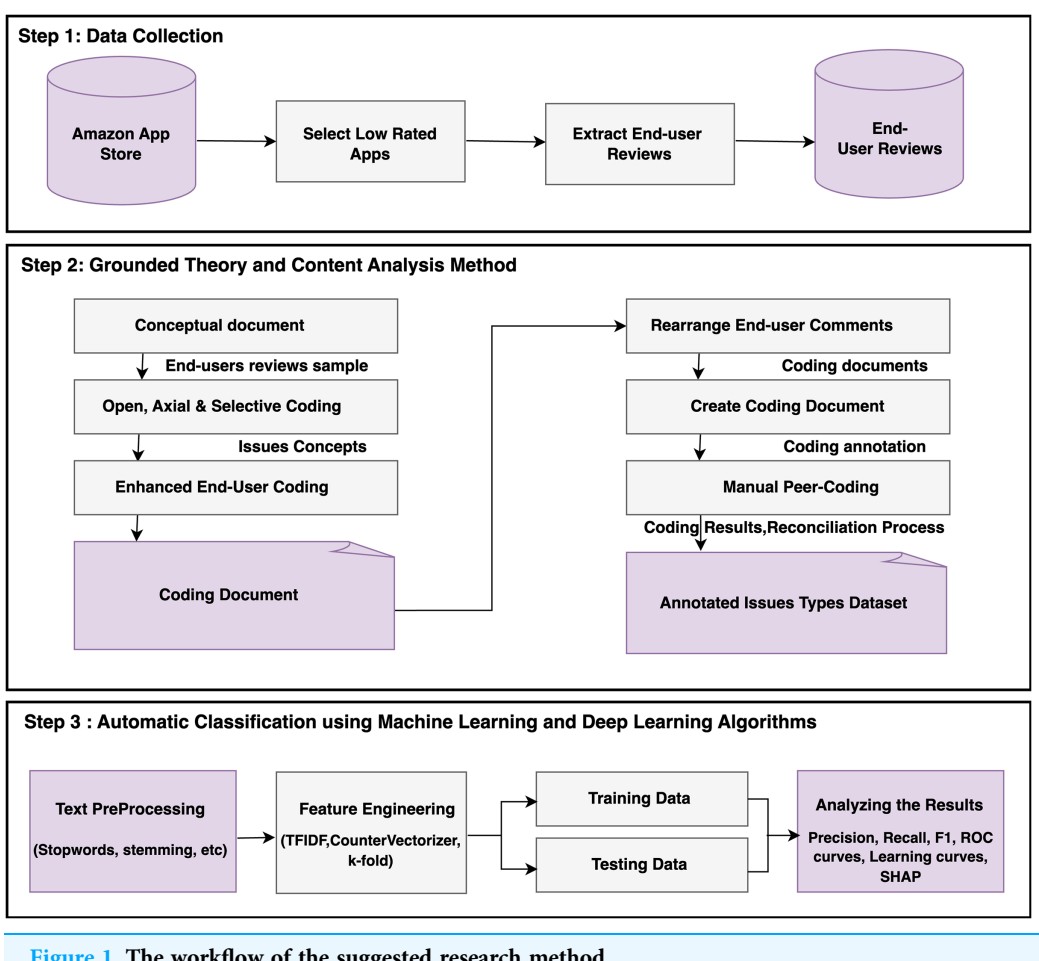

**Figure 1** The workflow of the suggested research method.

approach. This employs various NLP and feature engineering approaches to improve the performance of ML and DL classifiers. Additionally, it entails the proposed approach to employ the explainability approach (SHAP) to improve the decision-making process for software vendors and developers like the black-box ML or DL approach.

## Research method

As illustrated in Fig. 1, the proposed research method includes four significant phases and goals; each methodological step is discussed below.

### Research data gathering and development

To process the proposed approach, we require a research dataset comprising end-user reviews representing mainly issues-related feedback about software applications in the ASA store. For this purpose, we focused on applications with lower ratings, mainly three stars or less, in the ASA store. We concentrated on the ASA store because app stores have been given a lot of concentrations compared to ASA. Therefore, we were interested in identifying and exploring requirements-related information from ASA stores that is comparatively less explored as compared to app stores. We selected 64 software

applications from 14 unique categories in the ASA store, giving access to various software categories to collect a more generic sample for the proposed approach. Our selection criteria were based on finding applications with many reported issues and confirming that each picked application contained at least 350 user reviews. For this, the app reviewers were first manually skimmed by the first two authors of the article to confirm their suitability for the proposed approach. This strategy was critical in determining the underlying reasons for the low ratings and overall user experience, especially regarding end-user feedback. We used the Instant Data Scraper (https://chrome.google.com/webstore/detail/instant-data-scraper/ofaokhiedipichpaobibbnahnkdoiiah) extension in Google Chrome for the data collection. This extension was crucial in enabling the frequent extraction of end-user reviews and their associated star ratings, allowing us to rapidly handle and analyze a vast volume of data. We gathered a large data set of 79,821 end-user reviews, as shown in Table 1. Each record in the data set is comprehensive, including the author's name, title, full review, and the user's star rating. This approach counters the standard methodologies commonly used in requirements engineering literature, which typically focuses on more popular applications (*Khan et al., 2019b*; *Khalid et al., 2014*; *Guzman & Maalej, 2014*; *Khalid, Nagappan & Hassan, 2015*; *Yuan et al., 2015*; *Mezouar, Zhang & Zou, 2018*). Focusing on lower-rated applications uncovered helpful insights previously missed in distinct studies. Collecting this data into an exhaustive data set, as described in Table 1, provides a full view of the user experience with software applications on the ASA store. It includes positive, negative, and neutral user feedback, resulting in a more complicated and holistic view of user feedback in software applications. This extensive data set provides information on user preferences and issues. It inspires general trends and patterns in user feedback, which is essential for improving software quality and user satisfaction. However, in this article, we are interested in working with end-user reviews that represent potential issues regarding software applications.

### Grounded theory approach

In the second phase of the proposed research methodology, a pivotal step involves developing a coding guideline document. It is achieved by meticulously analyzing and evaluating a test sample of crowd-user comments. A random sample of 150 end-user comments from the distant categories of software applications in the dataset is selected manually. This process aligns with *Strauss & Corbin (1990)*'s Grounded Theory approach, a method known for its systematic and qualitative capabilities in theory building based on data evidence. Its purpose is to develop a novel theory from data samples by systematically analyzing them through comparative analysis, as evident in various studies (*Ullah et al., 2023*; *Chun Tie, Birks & Francis, 2019*). This phase results in the formation of a coding guidelines document, which serves as the final output, encapsulating the theoretical framework of the approach. As shown in Fig. 1, it is a critical artefact of the reconciliation step in the research process. The grounded theory approach, especially in recent applications, has been instrumental in creating the truth set or labelled dataset during the content analysis phase. This methodology helps decrease disagreement among the coders when annotating datasets for ML and DL experiments, providing uniformity and accuracy

**Table 1 Summary of software applications and user reviews collected from ASA store.**

| S/No. | Apps category | Software applications | End-user review |
|---|---|---|---|
| 1 | Business apps | 1: Hammer print, 2: Sketch guru, 3: Sketch book, 4: Logo maker & logo creator | 11,034 |
| 2 | Communication apps | 1: Justalk, 2: Textme, 3: Skype 4: AddMeSnaps, 5: Free text | 5,922 |
| 3 | Education apps | 1: TED TV, 2: Caspers company, 3: World now, 4: Amazon silk: duolingo.com | 6,968 |
| 4 | Food and drinks apps | 1: Food network 2: FoodPlanner, 3: Italian recipes, 4: Cheftap: recipe clipper, planner | 5,521 |
| 5 | Game apps | 1: Mobile strike, 2: Drive car spider simulator, 3: Chapters:interactive stories, 4: Crazy animal selfie lenses, 5: Cradle of empires-match | 4,748 |
| 6 | Movies and TV apps | 1: AnthymTV–It's free cable, 2: Hauppauge myTV, 3: StreamTV, 4: Trashy movies channel, 5: TVPlayer–watch live TV & on-demand | 5,982 |
| 7 | Music and audio applications | 1: Karaoke party by redkaraoke, 2: Mp3 music download, 3: Red karaoke sing & record, 4: Voice changer, 5: KaraFun-karaoke & singing | 1,889 |
| 8 | Novelty applications | 1: Ghost radar: classic, 2: Xray scanner, 3: Cast manager, 4: Age scanner, 5: Clock in Motion | 3,104 |
| 9 | Photos and videos apps | 1: AirBeamTV screen mirroring receiver, 2: AirScreen, 3: Snappy photo filter and stickers, 4: ScreenCast, 5: RecMe free screen recorder | 5,089 |
| 10 | Productivity apps | 1: PDF max pro–read, 2: Floor plan creator, 3: OfficeSuite free | 5,421 |
| 11 | Sports and exercise applications | 1: FOX sports: Stream live NASCAR, boxing, 2: FuboTV-watch live sports, TV shows, movies & news, 3: NBC sports, 4: CBS sports stream & watch live | 10,630 |
| 12 | Utilities apps | 1:PrintBot, 2: Floor plan creator, 3: TinyCam monitor free, 4: Tv screen mirroring, 5: Optimizer & trash cleaner tool for kindle fire | 3,689 |
| 13 | Lifestyle apps | 1: Screen mirroring, 2: DOGTV, 3: Fanmio boxing, 4: 3d Home designs layouts, 5: Home design 3D-free | 4,303 |
| 14 | Travel apps | 1: World Webcams, 2: Compass 3: Tizi world my pretend play town for kids 4: My route planner travel 5: Public transport maps offline | 5,521 |
| Total apps | 64 | | 79,821 |

in data understanding (*Strauss & Corbin, 1998*; *Neuendorf, 2017*). The developed coding guideline provides clear definitions and examples for each issue concept and instructions for labelling end-user comments in the data set. For the proposed approach, six frequently occurring issues concepts have been identified, including compatibility & device, functionality and feature, customer Support & responsiveness, security and privacy, UI and UX, and performance & stability by critically analyzing the test sample of end-user feedback. The detailed description of each issue concept is elaborated in "User-Defined Issues and Their Types" and the coding guideline (https://github.com/nekdil566/Comprehensive-Analysis-of-User-define-issues-in-end-user-reviews-/tree/main). Moreover, the first three authors of the article iteratively refine this guideline through discussions and reconciliation, reflecting the principles and challenges of grounded theory research to make it consistent for potential coders.

### Manual content analysis

The next phase in the proposed methodology is manual content analysis, involving a detailed manual annotation of crowd-user comments from a dataset. This task was undertaken by a dedicated team of human coders (the first three authors of the article) using the developed coding guideline and content analysis approach outlined by

*Neuendorf (2017)* and *Maalej & Robillard (2013)*. This step aims to address RQ2, focusing on extracting and identifying potential issues and their types within the user comments submitted against software applications on the ASA store. This step is crucial in forming the truth (labelled) sets essential for addressing RQ3, which revolves around the automation of the proposed methodology. The content analysis phase encompassed several critical tasks to ensure precision and reliability in the results. The initial step involved developing a stratified random sample of 8,971 end-user comments containing software issues or bugs for the various low-ranked software applications. These comments were categorized based on the specific software application they related to, as described in Table 1. This sample represented 11.24% of the total dataset, providing a significant and expected data collection size for analysis. Following the annotation process, we developed a structured coding form created in Microsoft Excel to facilitate the association and labelling of the 8,971 selected crowd-user comments that contain software issues or bugs. Along the coding guideline document, the coding artifact is distributed amongst the human coders to annotate the end-user feedback with the issue types identified and elaborated in the coding guideline document.

Each coder is asked to annotate the user feedback within the provided coding form, providing the developed coding guideline to assist them in annotating the feedback. After individually completing the annotation of end-user comments, the annotation results are compiled and integrated to resolve ambiguities if there are any. It will help to identify the accuracy and reliability of the annotations performed by the coders which will reflect the performance of ML and DL classifiers. We utilised statistical standards, such as the inter-coder agreement and Cohen's kappa (*Cohen, 1968*), to quantify the consistency and objectivity of the annotations across different coders. The coders took 25 working hours on average to complete the annotation process. During the procedure, conflicts among developers inherently arose. These problems were handled methodically through reconciliation and dialogue. This collaborative approach allowed us to manage disputes and differences, resulting in a blended, conflict-free labelled dataset. The inter-coding conflicts between the coders were calculated to be 88%. At the same time, Cohen's kappa was conveyed to be 68%, meaning that the annotators decided on the Cohen's kappa scale. The explanation of these conflicts was critical to the dataset's integrity and gave vital insights into the authentication of content analysis. Also, manual content analysis is a complicated and systematic process that includes several steps of sampling, annotation, and reconciliation. Each step was carefully designed and carried out to ensure the quality and reliability of the final labelled dataset, which is required for the next steps of the proposed research.

### Automated classification

In the final step, the process concentrated on evaluating the efficacy of various ML and DL classifiers in automatically classifying end-user reviews into various issue types. This procedure applies a series of research steps, for example, pre-processing techniques such as the NLP Toolkit2 and Stanford Parser3 are applied to the end-user feedback to clean crowd-user comments by removing stop words, special characters, images, HTML codes,

*etc*. Also, a steaming technique is applied that rounds the literary words to their initial form. Following pre-processing, feature engineering is applied to make the data pursuable and usable for the fine-tuned ML & DL classifiers, which entails converting the textual input into a numerical format that meets the needs of ML and DL algorithms. To do this, we used techniques such as TF-IDF and CountVectorizer, which are applied directly to processed text to extract and compute defining information, resulting in a comprehensive features matrix. In parallel, we employed the Label Encoder to convert non-numerical labels into a numerical format, allowing for automated classification.

Moreover, the ML & DL models are fine-tuned to automatically recognise and capture various issue types, including performance and stability, UI and UX, functionality and features, compatibility and device-specific, customer support and responsiveness, and security and privacy issues, which had earlier been identified manually for software applications with low ratings in the ASA store. We used a traditional cross-validation approach to refine and validate the ML and DL classifiers to confirm their robustness and accuracy on the classification task. Additionally, we employed various ML & DL evaluation matrices, including the receiver operating characteristic (ROC) curve, confusion matrix, precision, recall, and F1-score to validate the performance and generalization of ML and DL algorithms. Also, we used the Shapley Additive explanations (SHAP) approach to enhance the explainability of the classifiers which would help software vendors in understanding the results produced by the classifiers.

# PROCESSING END-USER FEEDBACK FOR ML AND DL CLASSIFIERS

This section elaborates on various frequently occurring issues along with examples identified in the end-user feedback in the ASA store. It helps annotate the end-user feedback, which is preliminary for automatically identifying issue-related concepts using ML and DL classifiers. Below, we elaborated in detail on the steps involved in the process.

## User-defined issues and their types

Nowadays, end-user feedback plays a vital role in installing or purchasing a software application (*Khan et al., 2023a*). End-user changes their purchasing or installation decisions while reading feedback for a particular software application (*Ullah et al., 2023*). Therefore, it is important to identify frequently occurring issues with the apps to retain customers and increase the pool of users. For this purpose, the proposed approach aims to meticulously extract and analyze end-user feedback that discusses critical issues with the apps. When analyzing the end-user feedback manually, we identified that end-users report issues that we grouped into certain user types, including performance and stability, UI and UX, functionality and features, compatibility and device-specific, customer support and responsiveness, and security and privacy issues. This research explicitly targets software applications with low star ratings, aiming to assess the impact of these issues on their overall quality and help software vendors identify frequently occurring issues to improve user experiences by including this information in the software evolution process. This proactive stance is expected to influence future updates, emphasizing user-requested

features and improvements in subsequent releases of the software applications. To achieve this, a comprehensive analysis and evaluation of end-user reviews on the ASA Store were conducted. This process involved identifying and labelling various user-experienced issues or bug elements. The annotation process played a pivotal role in this analysis, allowing for the emergence of specific issue types into well-defined concepts, thereby contributing to developing a grounded theory in software engineering for identifying issue types. The coding guideline included annotation codes based on how often they appeared in the user reviews and how relevant they were to the software issues. In the annotation process, codes that were either infrequent or unrelated to the research goals were either merged with the existing types or removed due to their low occurrence. This purified annotation process identified specific issue categories for low-ranked software applications in the ASA Store, containing performance and stability, UI/UX, functionality and features, compatibility, device-specific issues, customer support and responsiveness, and security and privacy issues. Each category has been elaborated upon with detailed definitions and extended examples, providing a comprehensive framework for understanding and addressing the diverse issues encountered by users of these software applications.

## Performance and stability issues

The "Performance and Stability Issues" code is assigned to the end-user feedback in the research dataset collected from the ASA store, demonstrating app crashes, freezes, or slowness, which risks the software functionalities and user confidence in continue using the software application. While manually analyzing the end-user feedback, it is observed that users frequently reported issues related to the app crashing, some functionalities becoming unresponsive, or comparatively slow performance of the app at times. It becomes a potential candidate for the low rating of the software application in the ASA store, resulting in the uninstalling of the app. Also, such feedback encourages other potential end-users to refrain from installing this app on their devices. For example, users report, "The app crashes continuously, rendering it utterly useless for any practical purpose." This statement indicates stability issues, which might deteriorate user confidence. Another user's detailed comment, "The app's performance is dreadfully slow, with loading times so extensive that it disrupts the entire workflow," falls into the same category. This input is crucial since it directly impacts the user's productivity and satisfaction with the app. These performance issues are not minor problems but significant blockages that can significantly impact the user experience and perception of the software. Therefore, it is essential to identify such critical issues and include them in the software evolution process to help improve software functionalities and end-user satisfaction.

## UI/UX issues

The code "UI and UX issue" is assigned to reviews in which the end-user expresses concerns over the app's layout and user interaction issues. We consider end-user feedback into this category that stresses the app's design and navigability issues. We argue that identifying such feedback is crucial for improving the overall app rating and user satisfaction. In contrast, app users become frustrated, resulting in or installing alternative

applications having better UI and UX. For example, a user in the App Store reports that "Navigating through the app feels like a maze, with a complex and unintuitive interface that leaves me frustrated," clarifies the critical aspects of user interface design that can make or break the user experience. This feedback is invaluable, providing direct insights into user efforts and dissatisfactions with the app's design. Another user expressed, "The visual design of the app is cluttered and visually overwhelming, detracting significantly from the user experience," further strengthening the need for a clean, user-friendly design. These insights are critical for understanding the user's point of view and for driving the design changes required to provide a more engaging and gratifying user experience. Therefore, bringing such issues upfront to the development team will enhance the UI and UX of the software applications, resulting in gaining end-user trust.

## Functionality and features issues

The code "Functionality and Features issues" is assigned to the end-user feedback, highlighting non-existent or dysfunctional features in the software applications. This issue category comprises user feedback containing information about missing, wrong, or incomplete software features that end-users are interested in. Similarly, these are core functional requirements of the potential end-users, and if missing or incomplete, they will cause them to switch to alternative software or even lead to uninstallation due to high frustration. For example, one user expressed, "The primary feature that was the reason for my download does not function at all, making the app redundant for my needs." This type of feedback is concerning because it negates the app's fundamental value proposition. Another user reported, "Several advertised features are conspicuously missing upon use, leading to a disappointing experience." This level of detail is required for developers to comprehend the gap between user expectations and the actual app experience. Addressing these gaps is critical to ensuring the app delivers on its promises and effectively satisfies user needs, resulting in high user ratings. Moreover, during the manual annotation of the end-user feedback, it is identified as one of the most frequently occurring categories, showing its importance towards achieving high quality and end-user satisfaction.

## Compatibility and device-specific issues

While analyzing the end-user feedback in the ASA store, we confronted reviews emphasizing an important non-functional aspect of compatibility issues. Therefore, the code "Compatibility and Device-Specific Issues" is assigned to end-user feedback highlighting problems related to the app's operation across various devices and operating systems. We argue that addressing compatibility issues is essential as end-users usually run the same software on multiple devices, *i.e.*, mobile and tablet or laptop, causing problems that will result in frustration. For example, the user reported, "The app exhibits flawless functionality on my smartphone but is completely dysfunctional on my tablet," indicating critical cross-device functionality issues. Another user elaborates, "Post the recent update, the app has ceased to work on my older Android device," pointing out the challenges in maintaining app compatibility across different versions of operating systems. These insights are crucial for developers to ensure that the app provides a consistent and reliable

experience across various devices and platforms. During the annotation process, it was observed that end-users usually report issues related to compatibility. It becomes crucial to regularly highlight such issues to the development team to improve or overcome such issues in the forthcoming versions of the applications.

## Customer support and responsiveness issues

When manually annotating the feedback in the test sample, we identified end-user reviews in which they complain about previously submitted reviews or frequently occurring issues that have been ignored or answered previously. They also complained about the customer support team, which is responsible for listening to the crowd confronting problems with the software application. We believe that such information is essential to bring upfront for the software vendors and developers to improve the software quality and enhance the customer support system. Therefore, we assigned the code "Customer Support and Responsiveness issues" to the end-user feedback that focuses on information related to customer support. It is essential to listen to the crowd and resolve their genuine problems with the software application, which will result in the low feedback collection necessary for the app's success (*Ali Khan et al., 2020*; *Seifert, Kuehnel & Sackmann, 2023*). For example, an end-user submitted a review, "My multiple attempts to reach customer support have been met with a deafening silence, leaving my issues unresolved," highlighting the value of responsive and effective customer care in the app ecosystem. Another user expressed displeasure, "The responses from customer support were generic and unhelpful, failing to address the specific problems I faced with the app," highlighting the importance of individualized and solution-oriented customer service. This feedback area is critical since it demonstrates the app's dedication to user pleasure and response to user wants and difficulties. Unlikely, if not listened to, the crowd will result in low-rating and user satisfaction. Moreover, gamification approaches will be unaffected, causing app ratings to suffer further.

## Security and privacy issues

Security and privacy concerns remain one of the most important non-functional concerns of end-users about software applications (*Ebrahimi, Tushev & Mahmoud, 2021*). While annotating the end-user feedback, we were confronted with reviews that demanded security and privacy concerns, mainly users' personal information. For this purpose, the code "Privacy and Security issues" is assigned to user reviews that express concerns about the app's handling of personal data and overall security perspectives. Identifying such problems is pivotal, resulting in distrust, limiting users' use of the application, and even recommending it to others. For example, a user alarmingly notes, "The amount of personal data the app requires is disconcerting and raises serious privacy concerns," highlighting the growing user awareness and concern over data privacy. Another review states, "I have noticed unusual activity on my account since installing the app, which makes me question its security protocols," identifying potential security exposures that can have far-reaching implications on user trust and app credibility. This form of feedback is vital because it has a direct impact on the user's perception of security and trust in the app, both of which are

essential for the app's success and retention. Furthermore, while analysing end-user feedback, security and privacy-related evaluations were the least common of the concern types. Nonetheless, it is critical to examine such challenges in the software evolution cycle to earn end-user trust and increase user happiness.

## Labelling user-expressed issues

To automatically classify end-user comments into frequently recurring issue types using ML and DL classifiers, we need to provide an annotated dataset, which is required. For this purpose, the content analysis approach (*Neuendorf, 2017*) annotates end-user feedback related to frequently occurring issues in the ASA store. The process entailed a comprehensive study of each user comment to categorize it based on the many issue types such as Performance & stability, UI & UX, functionality & feature-related, compatibility & device-specific, customer service & responsiveness, and security & privacy issues. The annotation method tries to identify the frequencies of distinct issue types to better understand their implications and the experimental setting by determining the number of instances in each category. Also, we develop a truth set essential for training and validating ML and DL classifiers and making the data publicly available for research purposes. This phase is critical in answering the study's RQ3 as if the annotation process is accurate and generalized resulting in better performances of various ML and DL classifiers. The first three authors of the research paper examined each user's feedback within the dataset to identify the concerns raised by the end-users about the apps in their feedback. It was decided to devise a comprehensive coding guideline to reduce the possibility of confusion and misunderstanding among the various coders. This guideline was critical in reducing conflicts between the coders. It was a comprehensive coding guide containing detailed definitions of each issue type along with examples. The guide served as a road map for the authors, ensuring that each comment was examined and classified objectively and consistently (*Khan et al., 2023a*).

## Annotation process for user-generated issue types

Annotating end-user feedback from the ASA store involves a meticulous and structured approach. Each coder is provided with a complete coding guideline and a coding form, including a set of potential issue types mentioned in user comments about software apps. These types include Performance & stability, UI & UX, functionality & feature-related, compatibility & device-specific, customer service & responsiveness, and security & privacy issues. The coding process begins with individually assessing each user comment's title and main content. Coders are tasked with identifying the associated issue type with each end-user comment. The coding form organizes the data efficiently for the annotation process. It includes columns such as "End-user Name," "Review Rating," "Review Title," and "Full Review," which present the user feedback information. Table 2 demonstrates instances of frequently occurring issues along their types from the dataset. A "UserIssue Type" column is included to categorize the potential issue type the coder identifies. For a structured and organized annotation, the end-user feedback is grouped by software application within the coding form. This design enables developers to easily access the original end-user reviews

**Table 2 Shows our manually labeled data set.**

| Name | Star | Review title | Full review | Issue type |
|---|---|---|---|---|
| Aden | 1.0 | Canon MG7520 not sup- ported. | I have a Canon Pixma MG7520 and this app will not work with this printer. When I sent my error message to them, this is the response: "this happens because the printer is not supported. Unfortunately, we are not sure when or if we are going to be able to create a mobile driver for this printer. There are no drivers that are known to be compatible with this printer" | Compatibility and device |
| Arman fatah | 1.0 | The login cannot work in fire HD 8 | This product is not working well on Amazon Fire HD 8. I am still trying to log in, and it cannot work. I used Many other Android phones and my account works Well. I called both Amazon and Skype support, but They didn't solve the issue | Customer support and responsive-ness |
| User3420 | 1.0 | Very dis- appointed app will work with Kindle Fire OS but is | Very disappointed app will work with Kindle fire OS but is very cumbersome. In order to perform a printing task, it is like you have to go through the procedure twice. The print options are very limited and 1 major problem it does not allow you to double side your output. Kindle OS really needs to be updated to something simple and useful for a simple task as printing from the device. very! very! disappointed in amazon! | Functionality and Features |
| Richard | 1.0 | Confusing | This may be ok for a tech wizard but I am a complete novice and found it confusing and all but impossible to delete unneeded files or even find and send a file I did need anywhere! | User inter- face and UX |
| Donna caissie | 1.0 | Doesn't Save | When I tried to save my spreadsheet, I found that the save icon was grayed out. Then I tried to email the spreadsheet to myself, but the email as attachment button was unresponsive. The quick send button was unresponsive too. | Performance and stability |
| Jon doh | 1.0 | Typically, invasive app | Like most other apps, they want to invade your privacy and security by demanding too many permissions with your phone. Personally, I refuse to allow that. So, I won't download any apps which want to invade my phone and steal information from it. | Security and privacy |

in situations of misunderstanding or uncertainty by providing direct links to each software application within the form. Comments about other software applications are set in order, followed by comments on a given application. Coders spend an average of 16 h annotating end-user comments. A reconciliation procedure is used to discover and analyse intercoder disputes after completing individual coding assignments. This stage is crucial for achieving a high level of coding agreement and verifying the validity of the annotated data. There was a high consensus among coders during this procedure, with a Cohen's kappa of 66%. This careful approach results in a dataset that is free of conflicts and labelled with the truth. In addition to serving as an input for ML and DL classifiers, this dataset is highly significant for software researchers and vendors in identifying useful insights for improving the quality of current app features.

## Frequency of frequently occurring issues types in ASA store

We employed content analysis methodology to thoroughly examine the end-users feedback and identify the frequency of issue types. This method is crucial for understanding end-user requirements and issues occurring against the software applications for software evolution. Also, it helps in setting various parameters, identifying features, and balancing data for the various ML and DL classifiers to generalize their results better in identifying frequently occurring issues. The analysis revealed that end-user

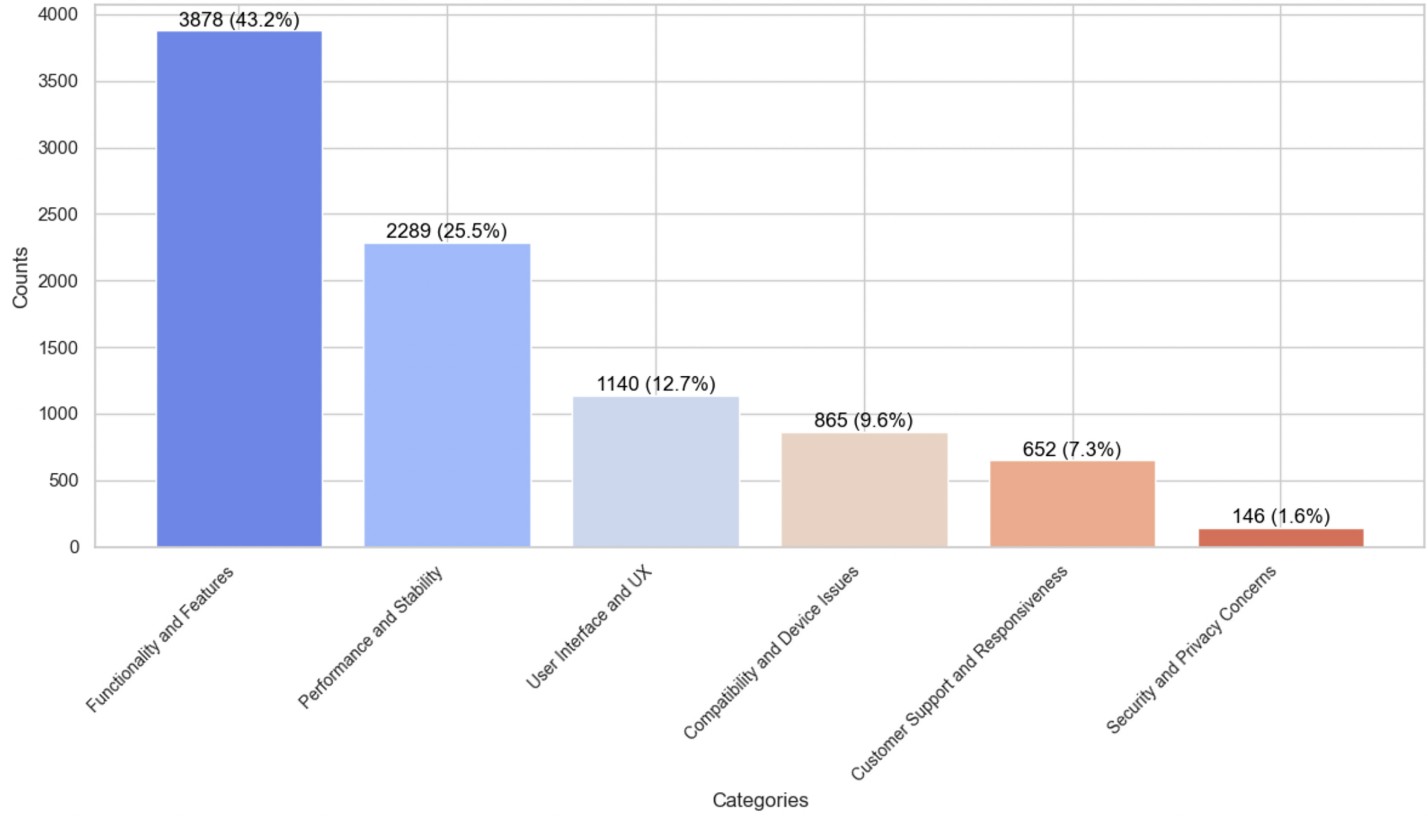

**Figure 2 Issues type distribution in the dataset.**

comments from the ASA Store are a valuable source for identifying frequently occurring issues for the software evolution and a possible source in improving the quality of existing applications by incorporating user concerns timely in the software evolution process. By the analysis, it is identified that functionality and feature-related issues are the most prevalent issues, representing 43.2% of the dataset (3,878 end-user comments), as demonstrated in Fig. 2. We argue that it was expected because we targeted software applications with comparatively low ratings and there is a possibility that apps earned low-rating due to missing or incomplete features. Therefore, the proposed study can work as a foundation for the software vendors and developers to consider such critical feedback in the software evolution process to improve the incomplete or missing features requested by end-users. The second most frequently occurring issue type is Performance and Stability-related end-user grudges with software applications, making up 25.5% of the dataset (2,289 comments). With the low-rating software application, it is expected that users complain about app performance, such as crashing when using the app, slow performance, *etc*. Such information can be a useful source for software developers to adhere to the key non-functional requirements during software evolution. Consequently, Compatibility and Device-related Issues were identified as the third most pressing issue type, accounting for 12.7% (1,140 comments) of the dataset, highlighting users' challenges with device

integration and software compatibility. Moreover, the study found that 9.6% of the comments (865 instances) were related to UI and UX issues. This emphasizes the impact of interface design on user satisfaction. Similarly, customer support and responsiveness issues were also notable, with 7.3% of comments (652) addressing this aspect. Lastly, security and privacy issues, while less systematic, were still significant, comprising 1.6% of the comments (146 instances). It might be because the software application selected for this study is not privacy-centric. Still, end-user are reporting their concerns about the app's security and privacy issues. This category indicates users' data security and privacy concerns while utilizing these applications. These are interesting findings for software vendors to consider for improving the quality of software applications. Also, these findings are crucial for requirements engineers, offering a comprehensive view of the users' challenges and problems. Such data is essential in developing more user-centred and practical software applications satisfying their needs and requirements.

## AUTOMATED CLASSIFICATION OF END-USER FEEDBACK INTO ISSUE TYPES

Recently, there has been a rising trend in publishing research articles highlighting the importance of end-user feedback in social media platforms for software evolution. However, these studies indicate substantial challenges in manual information processing for improved and timely decision-making to satisfy software needs and user satisfaction (*Khan et al., 2023a*; *Stanik, Haering & Maalej, 2019*; *Ullah et al., 2023*). Also, In the literature, most studies emphasize end-user feedback collected for popular and high-rating software applications, leaving a research gap to explore end-user feedback for low-rating software applications and recover opportunities for improving the app's ratings and user satisfaction. For this purpose, we collected 79,821 comments from the ASA store for 64 different low-rating software apps to explore the suitability of fine-tuned automated classification algorithms in identifying requirements-related information. To accurately reflect different user feedback, we methodically chose a subset of 8,971 reviews using a stratified random sampling method. The primary goal was to evaluate the effectiveness of various ML and DL algorithms in automatically classifying different issues highlighted in these end-user comments. We chose these ML and DL algorithms known for their better performance in extracting relevant information from textual data (*Ullah et al., 2023*; *Dąbrowski et al., 2022*; *Khan et al., 2023a*). The ML and DL classifiers selected for the proposed approach include MNB, SVM, LR, KNN, MLP classifier, gradient boosting, Voting classifier, random forest, Ensemble methods, NN, GRU, BiGRU, LSTM BiLSTM, and RNN.

Moreover, standard resampling techniques such as oversampling and undersampling were used to ensure a balanced dataset for generalized and accurate results, as shown in Fig. 2, which shows the distribution of comments among different issue labels. The ML and DL algorithms were trained and validated rigorously using the 10-K fold cross-validation method (*Kotsiantis, Kanellopoulos & Pintelas, 2006*), a frequently recommended approach in literature. This study aims to carefully examine all alternative algorithm configurations and uncover configurations that produce comparatively accurate and generalized

**Table 3 System configuration for ML/DL classification experiments.**

| Specification | Details |
| --- | --- |
| Computer | MacBook 2020 M1 |
| RAM | 8 GB |
| Processor | Apple M1 chip |
| Operating system | macOS 12.2.1 (21D62) |
| Python version | Latest version (3.12.1) |
| ML/DL libraries | Pandas, NumPy, Keras, scikit-learn, NLTK, Imblearn, Seaborn, Matplotlib, TQDM |

classification results. This strategy resulted in deploying numerous ML and DL models in varied configurations with commendably good precision, recall, and F1 scores. The article delves into the experimental setups, arrangements, and results of various ML and DL testing in-depth (*Khan et al., 2024a*; *Sarro et al., 2018*; *Kotsiantis, Kanellopoulos & Pintelas, 2006*).

## Experimental setup

The primary step in performing and conducting a classification experiment is to select ML and DL classifiers that perform comparatively better on the input data. For this purpose, we shortlisted MNB, SVM, LR, KNN, MLP classifier, gradient boosting, Voting classifier, random forest, Ensemble methods, NN, GRU, BiGRU, LSTM BiLSTM, and RNN ML and DL classifiers that produce better results with the short textual data (*Stanik, Haering & Maalej, 2019*; *Tizard et al., 2019*). The Voting classifier is a technique that combines the predictions of multiple classifiers to better generalize the classification results. The algorithm functions in two distinct modes: hard voting, which determines the final class based on the majority vote, and soft voting, which determines the final class based on the probability distribution across different classes. Ensemble methods, however, combine predictions from several classifiers, employing their collective strengths to improve accuracy. They commonly employ a blend of data partitioning and computational methods to achieve more significant outcomes. Also, each selected ML and DL classifier was trained and validated with end-user review to classify the input feedback to distant issue types, including performance & stability, UI & UX, functionality & feature-related, compatibility & device-specific, customer service & responsiveness, and security & privacy issues. Additionally, the ML and DL experiments were conducted in a Python-based environment, ensuring the strength and flexibility of the proposed study (*Ullah et al., 2023*; *Tizard et al., 2019*; *Khan et al., 2024a*). Table 3 shows the system configuration for the experiments.

## Preprocessing

In ML and DL research, preprocessing is a crucial step, especially when dealing with textual data, as it directly impacts the performance of the classifiers. A set of predefined operations ensures data integrity and quality during the preprocessing step. We first removed HTML tags and URLs detected in end-user comments to ensure the dataset

contains intelligible language for the ML and DL classifiers. We then transform the end-user feedback content to lowercase to standardize the dataset and eliminate discrepancies caused by case sensitivity. The process also involves removing extraneous characters such as brackets, punctuation, alphanumeric letters, and other special symbols that can affect the accuracy of the classifiers. The strategy enhances ML and DL algorithms by normalizing text using a technique known as lemmatization. For classifier accuracy to improve, lemmatization is essential for reducing words to their simplest form and maintaining regularity. Furthermore, removing stopwords is an essential part of preprocessing. To eliminate noise and focus on the dataset's more significant words, this step removes terms like "the," "is," and "in." To optimise data in ML and DL, algorithms need these insights to better understand and analyse the underlying logical difficulties (*Ullah et al., 2023*; *Sarro et al., 2018*; *Khalid et al., 2014*).

## Feature engineering

Feature engineering is crucial in ML and DL experiments, especially with textual data collected from social media platforms such as app stores, user forums, ASA stores, and Twitter. The proposed approach emphasizes employing textual features that have proven efficient in analyzing short texts collected from social media platforms, as indicated in earlier studies (*Stanik, Haering & Maalej, 2019*; *Tabassum et al., 2023*). The proposed approach primarily utilizes two frequently used techniques: TF-IDF and Bag of Words (BOW). In literature, the classification capabilities of these approaches have been acknowledged as superior in software and requirements engineering (*Stanik, Haering & Maalej, 2019*). The BOW approach entails creating a complete dictionary from the terms in the *corpus*, followed by tallying the frequency of each word in the user feedback. The TF-IDF technique assumes that commonplace words in a *corpus* contribute less unique information to ML classifiers than less frequent terms. We employed the TfidfVectorizer() and CountVectorizer() functions from the Python scikit-learn module to create TF-IDF and BOW, respectively. In addition, we utilized the N-gram feature, which involves grouping the frequently appearing sequences of N tokens in user comments. This feature is typically employed in ML classifiers for text classification. We specified the "ngram_range" option of the TfidfVectorizer from 1 to 3. It allows us to focus on detecting frequent word patterns in end-user comments that could indicate different user-identified problems. For example, expressions such as "frequent app crashes" or "login issues" may used as a possible indicator for technical difficulties issue type. Similarly, "confusing navigation" or "insufficient customer support" could indicate a usability issue type. The found patterns can be essential indicators for classifying end-user feedback into various frequently occurring issue types. In addition, we employed the Label Encoder function from the "sklearn. preprocessing" module in Python to convert textual data into numerical words, making it easier for ML and ML algorithms to handle the input data (*Sarro et al., 2018*).

Furthermore, we finetuned the DL algorithms by manually changing the hyperparameter values to devise the most appropriate potential results to determine the performance of DL classifiers in classifying issue types (see Table 4). To improve their performance, we modified several hyperparameters. One modification was to test the

**Table 4 Hyperparameter values used in fine-tuning DL algorithms.**

| Hyperparameter | Value |
| --- | --- |
| Activation function | Softmax |
| Loss function | Categorical crossentropy |
| Optimizer | Adam() |
| Embedded layer | 100 |
| Drop out | 0.2 |
| Dense layer | 64 |
| Epochs | 10 |
| Maxlen | 100 |
| Max features | 2,000 |
| Learning rate | 0.001 |

embedding layer with various values for the "output dim" parameter. We tested the performance of the DL algorithms with values of 100, 64, and 32. The highest level of accuracy was achieved with a setting of 100. Also, the Adam and Root Mean Squared Propagation (RMSProp) DL optimizers were employed with each DL classifier to identify their impact on the model accuracy. The results demonstrate that Adam optimizer outperformed RMSProp in identifying various issue types. The possible reasons for generating better results with Adam optimizer compared to RMSProp, include, Adam optimizer performing better with fewer training instances, as it converges faster (*Kingma & Ba, 2014*). In the proposed dataset, there are fewer instances in certain issue categories, *i. e.*, the Security and Privacy issue category has only 146 instances. Therefore, Adam optimizer is best suited when we have fewer instances. However, it might result in overfitting as it converges quickly and generates comparatively better accuracy and to cater for it we used a regularization approach by introducing a dropout layer. Also, Adam has the capability of handling noisy data more efficiently compared to the RMSProp optimizer (*Kingma & Ba, 2014*). As mentioned previously, textual data from social media platforms usually contains more noise, thus, Adam optimizer is a better option to be used with the DL classifiers with confronting more noisy data.

Furthermore, the DL classifiers encountered overfitting with training with the issues dataset, representing a situation when the DL classifiers memorize the training data rather than learn how to generalize. Overfitting is a common problem in DL that can severely limit the effectiveness of models when applied to unfamiliar data. To overcome the effect of overfitting, we employed regularization, reducing classifier architecture complexity, and incorporating dropout layers into DL algorithms. Interestingly, the CNN, GRU, BiGRU, LSTM, BILSTM, and RNN classifiers, performance is better generalized when a dropout layer with a dropout rate of 0.2 was added compared to regularization and reducing model complexity. DL classifiers could extract issue types while reducing the effects of Overfitting. Moreover, to better generalize the DL classifier results, we used different values such as 0.2, 0.3, and 0.4 with dropout rate, however, we get improved generalizability with the 0.2 value. Earlier, the results were overfitted, *i.e.*, the model memorized the outputs rather than

generalization. Also, its role is crucial in improving the model generalizability by randomly dropping out 0.2 (20%) of neurons during the DL model training, which helps the models not to rely on the particular neurons. Rather, it encourages learning more generalizable and robust features related to the various issue types while training the DL classifiers. It helps in reducing the overfitting by generating more reliable and generalized results with the fine-tuned parameters. Therefore, by integrating dropout layers into the models, their regularization capabilities improved, enabling them to adapt to novel data and accurately extract pertinent features associated with issue types from end-user comments.

## Data imbalance

Imbalanced datasets present a substantial technical obstacle in supervised ML and DL algorithms (*Chawla et al., 2002*), which is defined by an unequal distribution of annotation classes within a dataset. The developed dataset analysis showed a significant imbalance in the distribution of end-user comments, as depicted in Fig. 2. Out of all the end-user comments, the plurality (43.2\%) were found to be functionality and features issues, whereas a comparatively very small percentage (1.6\%) were categorized as security and privacy issues. This imbalance in data samples can suffer the training of ML and DL classifiers, causing them to show bias towards the more commonly occurring classes and disregard the minority ones, which have a smaller number of instances. To address this issue, we used two widely used data balancing techniques in software literature: oversampling and under-sampling (*Chawla et al., 2002*). To improve the performance of the ML and DL models and assure more accurate predictions for minority data, these methods balance class distributions in the dataset. Oversampling is a non-heuristic technique, which balances the instances across the dataset by adding duplicated instances for the minority classes (*Chawla et al., 2002*; *ChatGPT, 2024*). In contrast, under-sampling, a non-heuristic approach, aims to achieve balance by eliminating samples from the majority class (*Kotsiantis, Kanellopoulos & Pintelas, 2006*). We employed the Receiver ROC and Precision-Recall curves to identify the optimal technique for balancing the training of ML and DL classifiers in the proposed experiments. Figure 3 exhibits the ROC curves for two chosen ML classifiers, MLP and RF, evaluated using oversampling and under-sampling methodologies. We selected these classifiers based on their better performance during testing in classifying crowd-user reviews into different issue types. Based on the findings, oversampling outperforms undersampling in classifying end-user feedback to various issue types, which is logical as the undersampling technique eliminates vital textual data instances from the *corpus* that might be better for generalizing the ML and DL classifier results (*Tizard et al., 2019*). Therefore, we experimented with Oversampling with each ML and DL classifier to obtain more reliable and generalised results in classifying user feedback into distant issue types.

## Assessment and training

The proposed approach uses stratified 10-fold cross-validation for training and validating various supervised ML and DL algorithms. In software engineering literature, cross-validation has been widely used for improved and more generalized results with

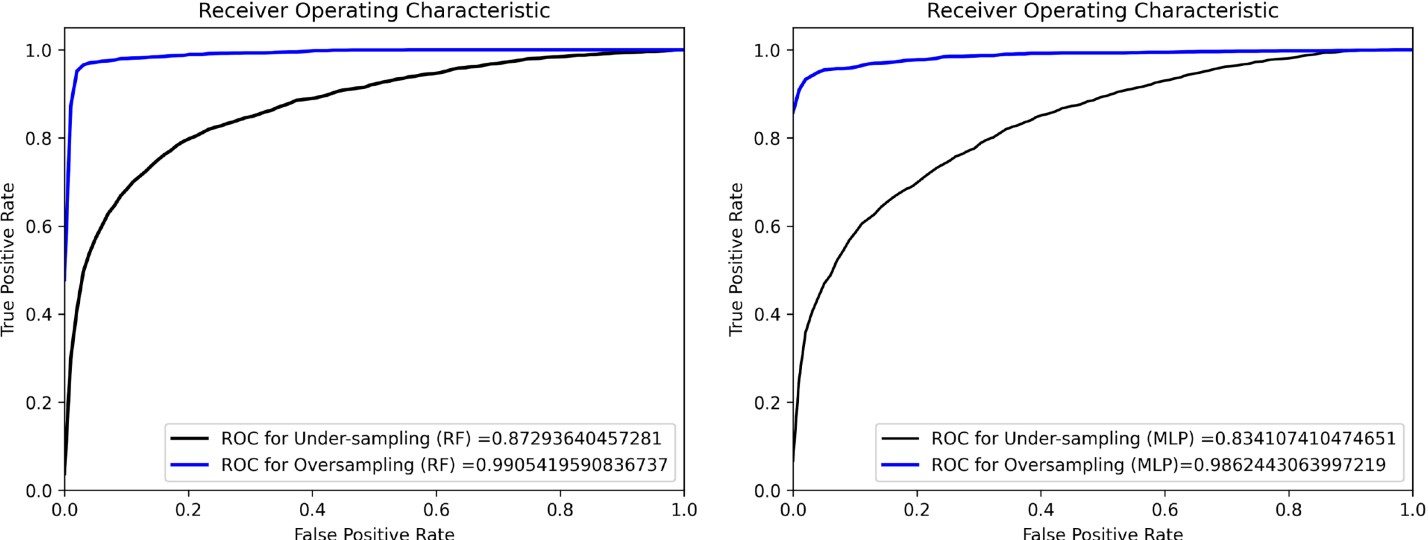

**Figure 3 ROC curves show the performance of oversampling and undersampling for RF and MLP classifiers.**

classification algorithms (*Ullah et al., 2023*; *Khan et al., 2023a*). For the cross-validation approach, 9 out of 10 folds are used to train ML or DL algorithms, while ten folds are used for validation. Each iteration of this training and validation process employs different training and testing folds. A stratified tenfold cross-validation technique is crucial to evaluate the model's performance thoroughly, especially when data are limited. This resampling technique enables a comprehensive evaluation of the model's performance using a limited dataset. To maintain the model's equilibrium and adaptability across different data sets, it ensures that each fold has an approximately equal distribution of all class labels. Based on tenfold cross-validation trials, we use a statistical technique to evaluate the classifications' performance. It is critical to analyze and assess the success of various supervised machine learning algorithms using Precision (P), Recall (R), and F1-score metrics.

$$Pk = \frac{TPk}{T_{Pk} + F_{Pk}} \tag{1}$$

$$Rk = \frac{TPk}{T_{pk} + F_{Nk}}. \tag{2}$$

P measures how well the model can correctly identify relevant end-user arguments (true positives) relative to the total number of comments classified, either correctly or incorrectly. In contrast, R assesses the model's ability to detect relevant examples from the overall number of instances present. It requires calculating the true positive rate of false negatives. The F1-score is a metric that combines Precision and Recall, providing a unified measure that represents the equilibrium between the precision and recall of the classifier. Balancing these two measurements is especially advantageous, offering a broader perspective on the model's performance.

## Results

The performance of various ML and DL algorithms in classifying end-user feedback into various issue types is shown in Table 5. We summarized the best-performing results of various ML and DL classifiers for classifying end-user feedback into various issue types, including compatibility and device, functionality and feature, customer Support and responsiveness, security and privacy, UI and UX, and performance and stability in Table 5. Comparatively, all algorithms performed effectively by classifying end-user feedback into compatibility and device issues. At the same time, CNN and GRU achieved the highest F-measures of 98% and 98%, respectively. Also, BiGRU excelled with a 97% F-measure, while BiLSTM showed a 92% F-measure when classifying compatibility and device issues. Similarly, GRU, BiGRU, and CNN maintained an impressive 98% F-measure when classifying end-user feedback into functionality and feature issues, but LSTM and BiLSTM lagged with 72% and 73% F-measures. One reason might be the kfold validation approach, which allocates various instances for the training and validation of ML and DL classifiers. In contrast, LSTM and BiLSTM marked a significant improvement in classifying end-user feedback into customer support and responsiveness issues, achieving a 95% F-measure, closely following RF-TFIDF, GRU, and CNN algorithms. Moreover, the LSTM and BiLSTM algorithms scored 98% and 99% in F-measure, outperforming other algorithms in security and privacy issues. Next, BiGRU, GRU, and CNN demonstrated outstanding results in UI and UX issues with a perfect F-measure of 99%, whereas LSTM and BiLSTM showed moderate results. For classifying end-user feedback into performance and stability issues, the BiGRU, GRU, and CNN classifiers scored the highest F-measures of 96%, 97%, and 96%, respectively. While LSTM and BiLSTM scored lower in classifying performance and stability issues. The overall average accuracy achieved by the classifiers revealed that MLP and RF stood at 94%, followed by BiGRU at 92%, GRU at 91%, CNN at 90%, and both LSTM and BiLSTM at 89%, indicating a competitive field with varying strengths across different issue categories. Interestingly, we observe that although MLP and RF algorithms achieved higher accuracy, DL classifiers such as CNN, BiGRU, and GRU produce better precision, recall, and F-measure values when classifying end-user feedback into various issue types. Upon investigating it, we found that DL classifiers, especially those utilizing recurrent neural networks like GRU and BiGRU, excel at capturing temporal dependencies and contextual nuances within the text, which often escape the capabilities of traditional ML models. They are particularly effective in text classification due to their layered, nonlinear architecture.

Furthermore, delving more deeply into the various model's performance examinations is essential to give researchers and software vendors a more detailed analysis in selecting the appropriate ML or DL classifier for the classification task. The MLP, RF, BIGRU, and GRU regularly outperform other ML and DL classifiers for classifying end-user feedback into various issue types. There are certain possible reasons for this when thoroughly analyzing the results obtained with these ML and DL classifiers. Firstly, as the dataset size is comparatively smaller in the proposed approach and GRUs classifiers use fewer parameters during training, making them computationally less intensive and potentially

**Table 5 The performance of ML and DL algorithms to classify user comments into issue types.**

| Issue | Algorithms and features | Precision | Recall | F-measure |
|---|---|---|---|---|
| Compatibility and device issues | MLP-CountVectorizer | 96 | 93 | 93 |
| | RF-TFIDF | 95 | 93 | 94 |
| | BiGRU | 96 | 99 | 97 |
| | GRU | 97 | 99 | 98 |
| | CNN | 98 | 98 | 98 |
| | LSTM | 95 | 96 | 96 |
| | BiLSTM | 88 | 97 | 92 |
| Functionality and features issues | MLP-CountVectorizer | 94 | 96 | 95 |
| | RF-TFIDF | 95 | 92 | 93 |
| | BiGRU | 98 | 99 | 98 |
| | GRU | 97 | 99 | 98 |
| | CNN | 98 | 98 | 98 |
| | LSTM | 85 | 62 | 72 |
| | BiLSTM | 80 | 67 | 73 |
| Customer support and responsiveness issues | MLP-CountVectorizer | 94 | 91 | 92 |
| | RF-TFIDF | 94 | 96 | 95 |
| | BiGRU | 96 | 93 | 94 |
| | GRU | 94 | 94 | 94 |
| | CNN | 94 | 96 | 95 |
| | LSTM | 92 | 99 | 95 |
| | BiLSTM | 91 | 99 | 95 |
| Security and privacy issues | MLP-CountVectorizer | 71 | 93 | 81 |
| | RF-TFIDF | 95 | 96 | 95 |
| | BiGRU | 95 | 98 | 96 |
| | GRU | 96 | 98 | 97 |
| | CNN | 96 | 98 | 97 |
| | LSTM | 98 | 98 | 98 |
| | BiLSTM | 97 | 99 | 99 |
| UI and UX issues | MLP-CountVectorizerv | 93 | 90 | 90 |
| | RF-TFIDF | 94 | 95 | 95 |
| | BiGRU | 99 | 99 | 99 |
| | GRU | 99 | 99 | 99 |
| | CNN | 99 | 99 | 99 |
| | LSTM | 81 | 93 | 87 |
| | BiLSTM | 89 | 89 | 89 |
| Performance and stability issues | MLP-TFIDF | 95 | 95 | 95 |
| | RF-TFIDF | 97 | 88 | 90 |
| | BiGRU | 99 | 94 | 96 |
| | GRU | 97 | 96 | 97 |
| | CNN | 99 | 92 | 96 |
| | LSTM | 83 | 86 | 85 |
| | BiLSTM | 89 | 84 | 87 |

| Table 5 (continued) | |
| --- | --- |
| ML and DL classifiers | Best average accuracy |
| Multi-layer perceptron classifier | 94 |
| Random forest classifier | 94 |
| BiGRU | 92 |
| GRU | 91 |
| CNN | 90 |
| LSTM | 89 |
| BiLSTM | 89 |

better choice compared to LSTMs algorithms with limited training data. Also, LSTMs classifiers are more exposed to overfitting because they have more parameters compared to GRUs, especially if the dataset is comparatively smaller. Similarly, LSTMs classifiers are more prone to the commonly occurring vanishing gradient issue during training compared to GRUs due to their additional gating procedure leading them to slower convergence and comparatively poor performance. Moreover, DL models naturally maintain the ability to extract complicated patterns and nuances from textual data due to their layered, non-linear architecture. This characteristic allows them to understand the text's temporal relationships and contextual subtleties, which traditional ML models may find difficult to understand. Besides, DL models ability to automatically discover and adapt to feature representations in large and diverse datasets gives them a distinct advantage in accurately categorizing a wide range of end-user feedback into specific issue types. Moreover, with their capacity to capture intricate patterns, DL algorithms can result in high precision, recall, and F-measure values, especially in complex tasks, surpassing traditional ML algorithms (*ChatGPT, 2024*), as shown in Table 5. This subtle experience highlights the significance of considering both the dataset's features and the inherent powers of DL models when analyzing performance metrics. Additionally, MLP and RF having a better accuracy compared to LSTMs might be that issues input textual data is well-structured due to the extensive pre-processing and feature engineering processes resulting in meaningful data presentation making it more suitable for MLP and RF classifiers. Also, the input data is not sequential and does not contain long-term dependencies, whereas, LSTMs classifiers are designed to capture long-term dependencies between the data. Therefore, MLP and RF results in better accuracy compared to LSTMs. Finally, the dataset size is comparatively smaller in the proposed approach, where LSTMs require a larger dataset to better generalize because of its many parameters. These are the possible reasons, based on our knowledge, that MLP, RF, and BiGRU, GRU consistently outperform BiLSTM and LSTM for classifying end-user feedback into various issue types.

In light of the detailed performance metrics in Table 5, the selected ML and DL classifiers have shown comparatively better proficiency in the multi-class classification of Amazon software reviews. These algorithms have shown superior performance, particularly in detecting compatibility, functionality, customer support, security, and UI/

UX issues, among which are MLP, Voting, GRUs, and RF classifiers. As well as showing high precision, recall, and F-measure values, these classifiers outperformed the competition in specific areas. LSTM and BiLSTM excelled in security and privacy, while CNN and GRU excelled in UI and UX. This slight performance highlights these algorithms' ability to capture subtleties in user feedback. Based on the results in Table 5, any of these classifiers can be used optimally to analyze diverse issues in user comments, a significant improvement over previous techniques.

Furthermore, we looked at the baseline configurations of the best classifiers, MLP and RF. As a central part of this analysis, we examined the learning curves of training datasets to see how size affects classification accuracy. Besides that, Figs. 4A–4D show how long it takes to train each classifier size. Based on its training efficiency and accuracy, the MLP classifier was considered as the best-performing classifier in classifying end-user comments into various issue types. Based on its learning curve, as shown in Fig. 4C, it demonstrated its superior ability to handle various issue types in user comments. In contrast, its efficiency in training time, shown in Fig. 4, demonstrated its suitability for issues mining applications, especially on digital platforms where user feedback is essential.

In addition, We also looked at GRU and BiGRU's training and validation loss and accuracy, as detailed in Fig. 5. It is important to determine how these classifiers learn and perform on new, unseen data since training loss measures how well the model fits the training data. Validation loss and accuracy are a measure of its generalization. Figures 6A and 6B shows the receiver operating characteristic (ROC) curve of the GRU and BiGRU classifiers. ROC curves compare the actual positive rate with the false positive rate to determine whether a classifier is sensitive and specific. By examining both loss and accuracy metrics, as well as the ROC curve analysis of the GRU and BiGRU classifiers, we can understand how well they classify text within user-defined issues, which is crucial to analyzing input from digital platforms.

Moreover, the Area Under the Curve (AUC) demonstrates comparatively better classification performance for the GRU and BiGRU models in classifying end-user feedback into various issue types. As a result of the high accuracy of the AUC values, each class achieves or approaches one value, as shown in Fig. 6. Actual Positive Rate (TPR) is high, and False Positive Rate (FPR) is low, as shown by the curves approaching the upper left corner. Both models have a macro-average AUC of 1.00, showing that they can classify all classes uniformly and consistently. TPR and FPR are calculated by formulae.

$$TPR = \frac{TP}{T_P + F_n} \tag{3}$$

$$FPR = \frac{FP}{T_n + F_p}. \tag{4}$$

These measures indicate the proficiency of both the GRU and BiGRU models in reliably recognising each class while minimising misclassifications, therefore verifying their effectiveness in multiclass classification contexts.

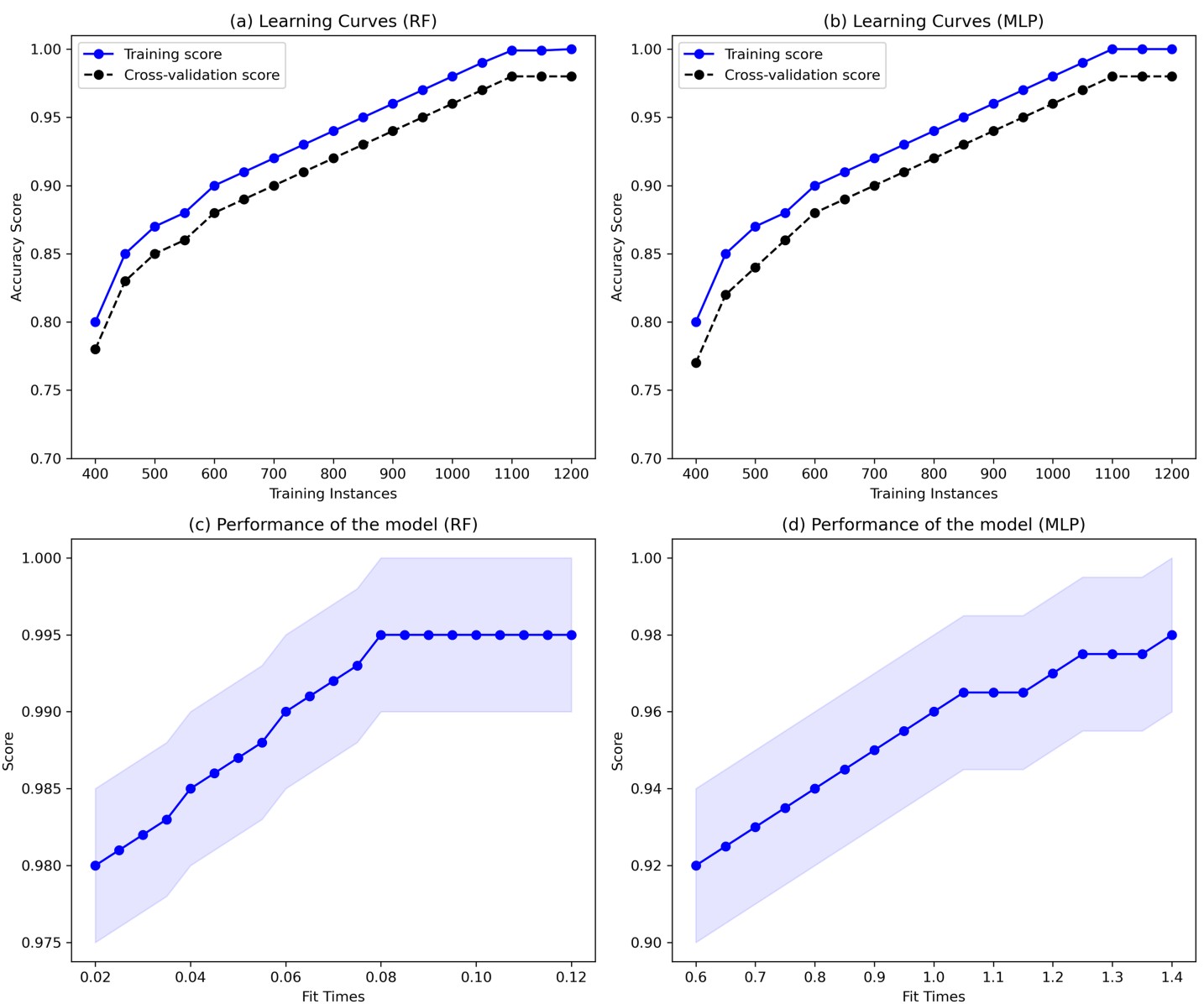

**Figure 4 Learning curves show the learning rate (A and B) for RF and MLP algorithms and (A and D) shows the training time (B and D) for ML classifiers.**

Furthermore, confusion matrixes for the GRU and BiGRU models illustrate how well they classify customer feedback categories, as shown in Figs. 7A and 7B. In the GRU matrix, many categories show high numbers of true positives, with 'Compatibility and Device Issues 'and' User Interface and UX' being classified with high accuracy, with 4,027 and 4,079 correct classifications, respectively. Across categories, misclassifications are relatively low, but 204 instances of 'Performance and Stability' feedback being misidentified as 'Functionality and Features' are notable. According to the BiGRU matrix, 'Compatibility and Device Issue' and 'User Interface and UX' both have high correct
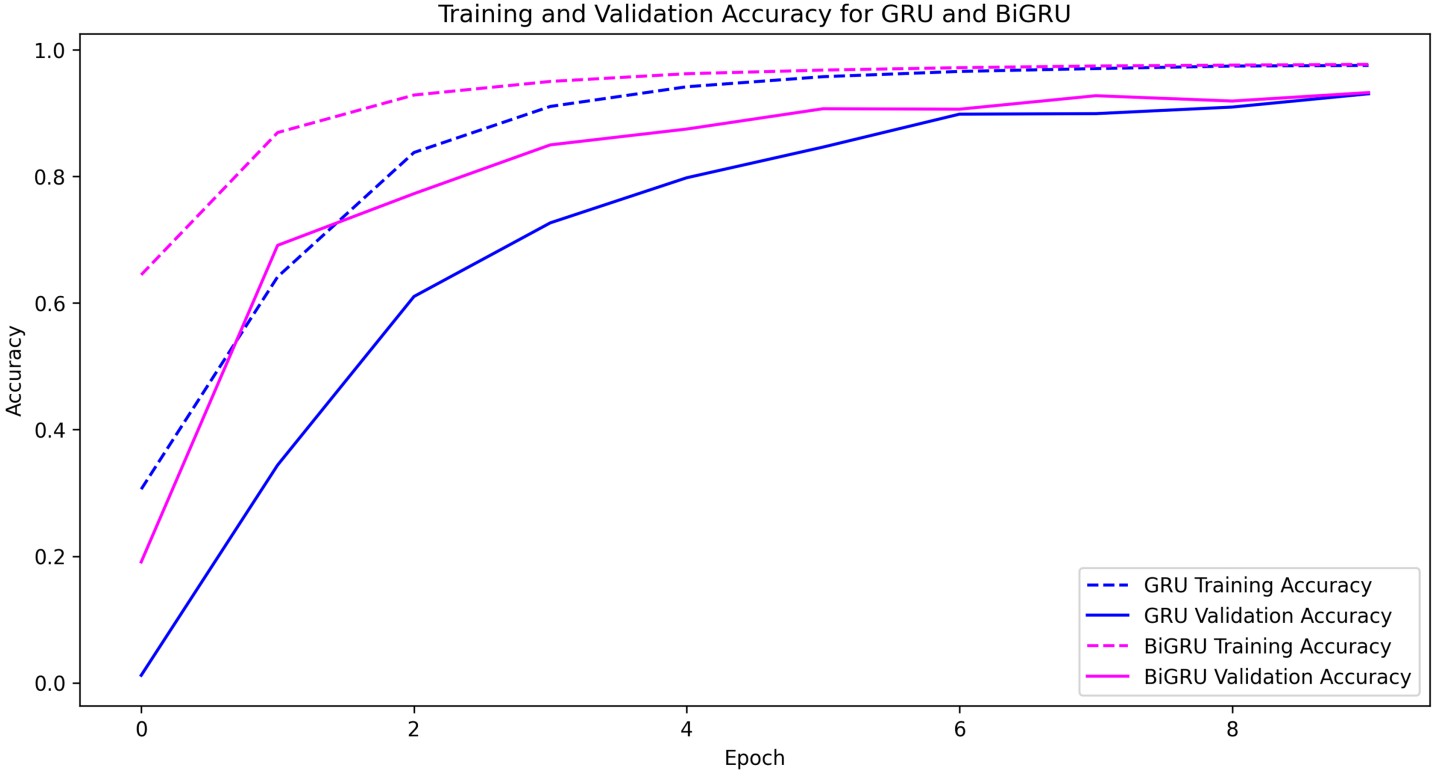

**Figure 5  Training and validation loss and accuracy of GRU and BiGRU classifier.**

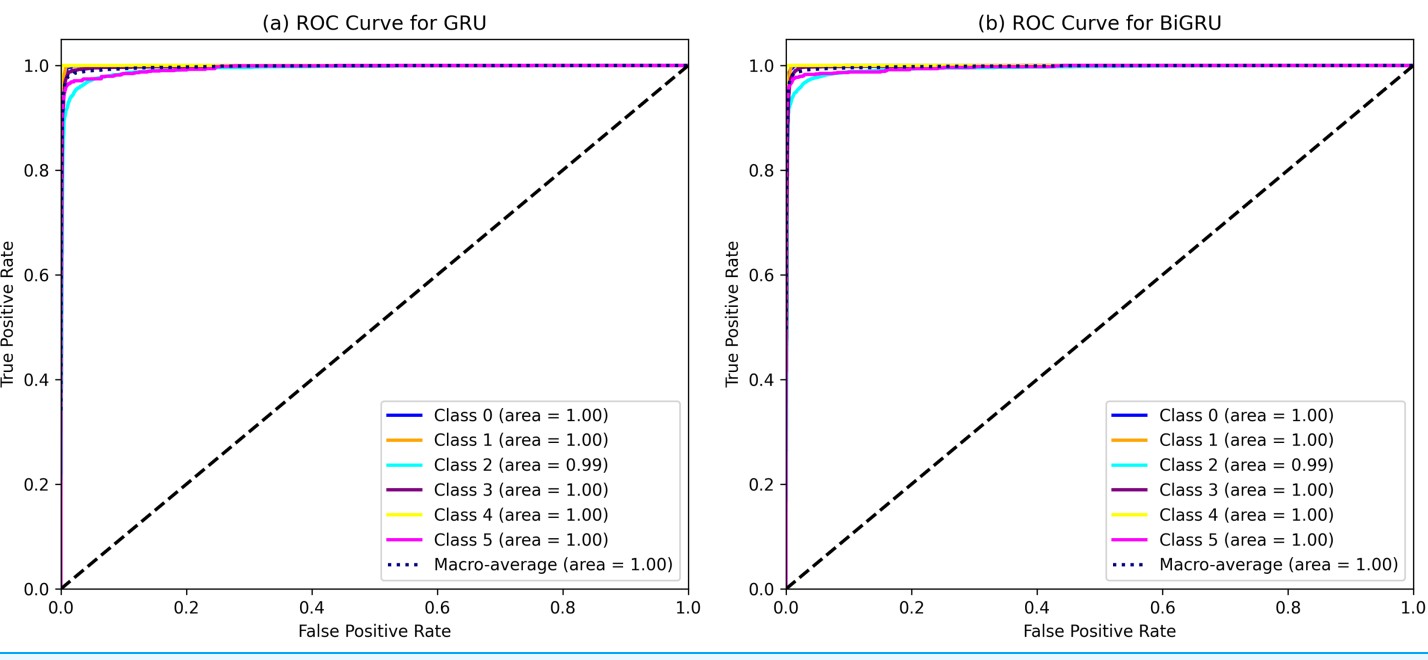

**Figure 6  ROC curves of GRU and BiGRU classifier.**
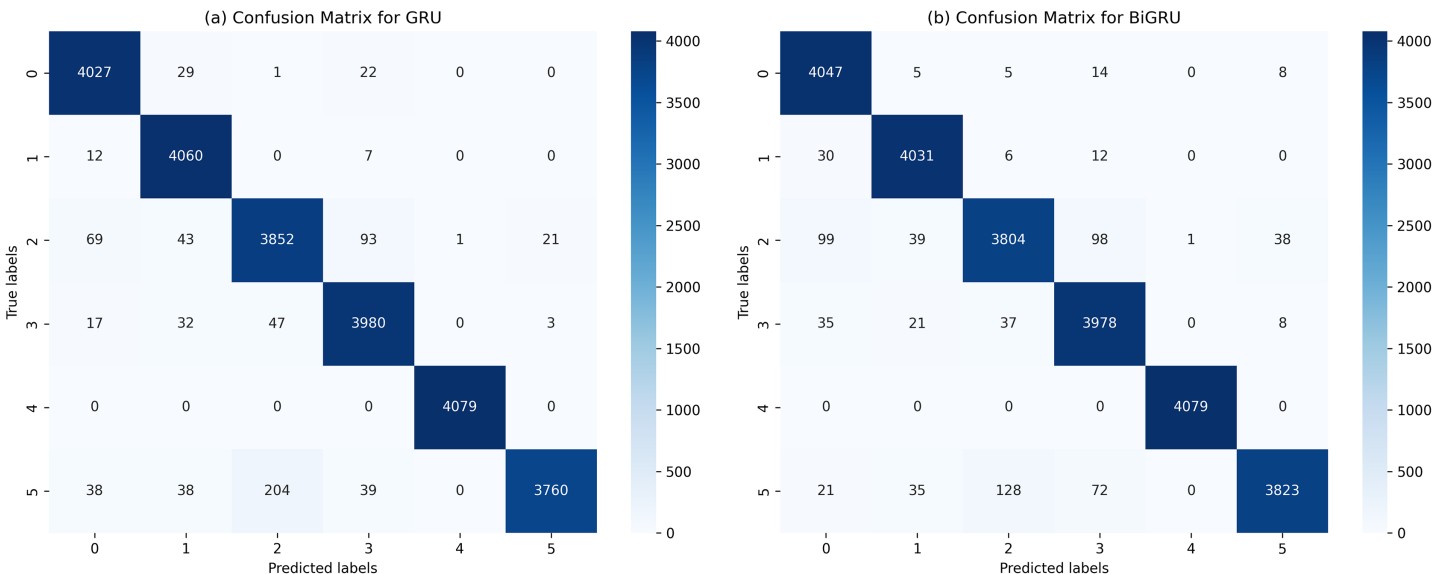

**Figure 7 Confusion matrix of GRU (A) and BiGRU (B) classifier.**

classifications of 4,031 and 4,079, respectively. For some issue types, misclassifications are higher in the BiGRU model, such as the 128 instances where 'Performance and Stability' feedback is misclassified as 'Functionality and Features' feedback. While both models effectively classify feedback across categories, they exhibit different strengths and weaknesses.

Recently, there has been a high demand in software engineering literature to improve the explainability of the proposed fine-tuned classification approaches to help software researchers and vendors understand the black-box nature of these classification algorithms to make better-informed decision-making. For this purpose, to perform a preliminary experimental study by examining the impact of various textual features on ML models, we used the SHAP (Shapley's Additive Explanations) method with 3,000 reviews from various categories. Also, the experiment was performed on MacBook Pro 2020 having computational constraints for taking larger dataset instances. Therefore, we used a random sample of 3,000 end-user feedback across different issue categories. These reviews were divided into six issue types: performance & stability, UI & UX, functionality & features, compatibility & device-specific, customer support & responsiveness, and security & privacy issues. In the analysis, detailed SHAP value graphs, presented in Figs. 8A–8F, showed the differential impact of particular terms within these categories. For instance, "crashes" and "slow" were critical terms in performance and stability. Figure 8B shows that words like "confusing" and references to specific apps like "Hulu" were significant for UI and UX. Figure 8C shows that terms like "missing" and "upgrade" stand out when it comes to functionality and features. In comparison, "compatible" and "iOS" played a significant role in compatibility and device issues, as shown in Fig. 8D. A keyword in customer support and responsiveness was "customer" and "cancelled", and a keyword in security

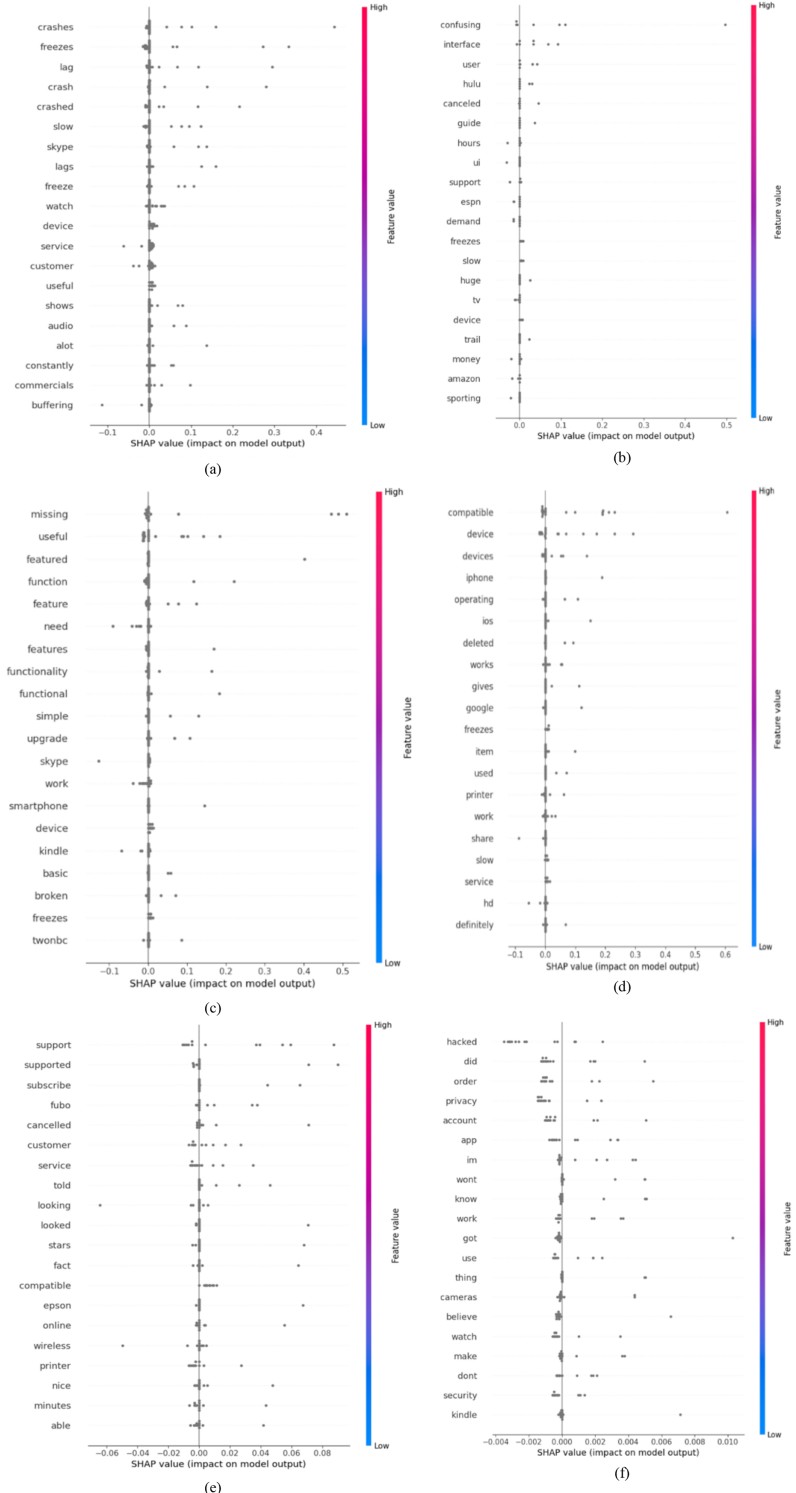

**Figure 8 Key terms influencing MLP classifier predictions on performance, UI/UX, features, compatibility, support, and security, *via* SHAP analysis.**

**Table 6 Comparative analysis of prior studies and proposed approach in issue detection in software reviews.**

| Approaches | Highest accuracy | Issue F-measure | Applied explainability | Issue types |
|---|---|---|---|---|
| *Ullah et al. (2023)* | 93 | 92 | yes | No |
| *Khan et al. (2022)* | N/A | 52 | No | No |
| *Alkadhi et al. (2018)* | N/A | 59 | No | No |
| *Kurtanović & Maalej (2018)* | N/A | 75 | No | No |
| *Maalej & Nabil (2015)* | N/A | 72 | No | No |
| Our proposed approach | 94 | 99 | Yes | Yes (UI and UX, functionality and features, compatibility and device-specific, customer support and responsiveness, security and privacy issues, performance and Stability issues) |

concerns was "privacy" and "security," as shown in Fig. 8F. These SHAP plots can help software researchers and vendors understand the reasons why a particular end-user feedback is classified into a particular issue type. It can help software development teams in understanding the frequently occurring issue types associated with the software applications in the ASA store. They can use this information to improve the overall quality of software apps and user satisfaction by focusing on the information portrayed in the SHAP plots, as shown in Fig. 8 during the software evolution. This experimental study highlights the keywords or features that reflect current user perceptions about software applications and demonstrates the need for a more integrated software evolution process to timely and efficiently incorporate the information identified in the improvement of software applications.

## Comparative study

As elaborated previously, in the proposed approach, we emphasise elaborating frequently issues in the ASA store for low-rating software apps, which we believe, to date, according to our knowledge is the first study in the requirements engineering literature. However, in Table 6, we provided a comparative study with the most close and research approaches that focus on extracting and analysing issue information from various social media platforms. For instance, *Ullah et al. (2023)* achieved comparatively better results for identifying various rationale-related concepts from the ASA store, achieving an overall highest accuracy of 93% and an F-measure of 92% for identifying issue rationale elements. Similarly, *Khan et al. (2022)* analyzed end-user comments from the Reddit forum to identify and classify various rationale elements including issues. They achieved a 0.52% F1 value with the issue rationale element. Similarly, *Alkadhi et al. (2018)* achieved a 0.59% F-measure value in classifying developers' discussions from specialized developers' forums,

*i.e.*, the Apache Lucene website to issue rationale elements. Similarly, *Kurtanović & Maalej (2018)* and *Maalej & Nabil (2015)* achieved comparatively better F-measure values of 75% and 72%, for classifying user feedback into possible issues from ASA and Mobile Paly stores (Google & Apple), respectively. In contrast, the proposed approach explores end-user feedback from the ASA store about low-rating software apps from a different perspective to classify them into frequently occurring issue types, such as UI and UX, functionality and features, compatibility and device-specific, customer support and responsiveness, and security and privacy issues. Moreover, using fine-tuned ML and DL classifiers, we achieved a comparatively better accuracy of 94% in classifying end-user feedback from the ASA store into various issue types. Moreover, we achieved the highest F-measure value for classifying end-user feedback into UI and UX and security and privacy issue types. Additionally, unlike other approaches (except *Ullah et al., 2023*), we used SHAP to identify keywords associated with each issue type. It would give a comprehensive overview to the vendors and developers in understanding associated concepts with each issue type and provide opportunities to improve app quality and user satisfaction by timely incorporating this information in the software evolution process.

## DISCUSSION

This study emphasizes the importance of user feedback in software evolution, particularly focusing on low-rating software applications by exploring frequently occurring issue types. The proposed approach supports the previous research claims on the usefulness of end-user feedback in the software evolution process from different viewpoints. This study highlights that end-user feedback serves as a vital resource for identifying and understanding the repeatedly occurring issues and challenges in lower-rated software applications in the ASA store. These user reviews are not simply comments but are rich sources of leveraged understandings, pouring light on the various frequently occurring bugs, usability issues, and software features that need improvements. Further, the data emphasizes that user feedback within the ASA store encompasses more than merely identifying software program flaws. The inclusion of this insightful knowledge in the software evolution process plays a crucial role in guiding software developers and vendors toward significant improvement by incorporating frequently occurring issues to improve user satisfaction and overall quality. Through meticulous examination of these examinations, developers might develop a deeper understanding of consumers' expectations, preferences, and dissatisfactions with software applications or certain software functionalities. Understanding this is crucial for future software updates and development to more effectively meet the requirements of users. This study prioritizes the importance of user evaluations in the context of software development within app marketplaces. Software developers can improve and promote their software applications through the adoption and detailed analysis of customer feedback. The ongoing process of feedback and improvement plays a vital role in securing the success and long-term viability of the program within the fiercely competitive app store market. In the following section, we explain the diverse views that can be attained with the proposed methodology.

## User feedback in software development

The literature has emphasized the significance of user feedback in software development, especially in the domain of app stores, as an absolute factor in improving software quality and encouraging innovation (*Khan et al., 2019a*; *Khan, Liu & Wen, 2020*; *Ali Khan et al., 2020*; *Ullah et al., 2023*). The outcomes underscore the significance of a detailed examination of user reviews, as they serve as vital sources of firsthand understanding of user needs and experiences. User feedback is a collection of user opinions from social media platforms and serves as a crucial instrument for the iterative enhancement of software applications. The venue offers designers access to up-to-date data to enhance and optimize their software applications. Similarly, the narrative is supported by the research undertaken by *Iacob & Harrison (2013)* and *Carreno & Winbladh (2013)*, which shows how user feedback can effectively identify a range of software-related issues, including bugs, feature requests, and more comprehensive user experience problems. Compatible with existing research works, the proposed methodology examines end-user reviews from a different perspective to analyze them by focusing on software-related concerns registered by the end-user against low-rating software apps. On the other hand, the feedback delivered by end-users is classified into many categories of issues, such as compatibility and device compatibility, functionality and features, customer support and response, security and privacy concerns, user interface and user experience, and performance and stability. These issue classifications hold significant value for software applications that are forced by market demands, as the success and durability of such systems are directly affected by user satisfaction. Developers can improve the usability of software developments and meet the varying needs and anticipations of their target audience by gathering, evaluating, and implementing user feedback. Likewise, the data gathered using the proposed methodology contains typically experienced critical problems of diverse kinds, which is crucial to communicate directly to software developers to improve the software applications by performing customer delight. Therefore, it is crucial to adopt a user-centric strategy towards software development and evolution within the current digital landscape. The need for a flexible and adaptive strategy of software creation and enhancement arises from the quick change in user choices and market trends.

## Analytical framework for software evolution

The proposed research study employs a comprehensive analytical methodology to assess user feedback for issue-related information. Also, in line with similar research studies (*Ullah et al., 2023*; *Maalej & Robillard, 2013*; *Strauss & Corbin, 1998*), we use content analysis and grounded theory techniques to systematically identify frequently occurring issue types resulting in a unique coding guideline and annotated dataset. Furthermore, researchers and software vendors who have shown the ability to derive pertinent insights from qualitative data, particularly in contexts like end-user feedback for software evolution, might benefit from the proposed approach. The proposed approach is aligned with the *Neuendorf (2017)* methodology recommended for identifying frequently occurring concepts related to software evolution. This methodological technique to organise and comprehend data categorisation requires a thorough comprehension of end-

user feedback. In the proposed approach, following the guidelines from *Neuendorf (2017)*, the end-user feedback is categorised and classified into performance concerns, user interface and experience complaints, and functionality issues. This systematic classification is used to help turn massive volumes of qualitative data into actionable insights for software evolution. The proposed approach allows us to move beyond defining analysis and into interpretive insights, ultimately resulting in a more in-depth understanding of user experiences and opinions about why users submit negative feedback and low ratings for the software apps on the ASA store. The theoretical framework of the proposed approach can be utilized by software researchers and vendors for similar-nature software problems. These techniques are crucial in app store assessments, including a large amount of user feedback to understand the reasons why users feel dissatisfied with the performance of the software apps.

## Implications of identified trends and patterns

With the spark of social media and trends of market-based software applications, software developers and vendors need to understand user feedback, as it enables them to gain insight into user preferences, encountered difficulties, and overall app experiences. Analyzing the emerging trends and patterns identified in user feedback by critically analyzing using manual and supervised approaches can provide software developers and vendors with valuable insights into improving the existing software apps and enhancing user satisfaction. Software applications can be enhanced based on these patterns, which is one of the notable advantages of recognizing them. However, it is mandatory to update the existing software evolution process to incorporate the frequently identifying software evolution-related information in the process for improved user satisfaction and quality. For example, developers may allocate resources towards improving the speed and reliability of apps if many users express dissatisfaction with the performance of the app features. Also, if end-users repeatedly complain about the interface or user experience of the apps, developers may redesign these features to improve the software's accessibility and user usability.

Moreover, developers can gain insights into the changing preferences of their intended demographic by analyzing user input. Developers can adjust their development plans according to users' changing expectations by observing changes in user sentiment over time. A proactive strategy may allow software apps to remain competitive and relevant in today's ever-changing app market. However, user input is essential to software development and broader organizational initiatives. A good example is when customer maintenance teams use user evaluations to resolve customer problems effectively, increasing customer satisfaction and encouraging commitment. Analyzing trends and patterns in user evaluations provides valuable information that can be used to enhance software programs, connect development objectives with changing user preferences, support compelling features, and eventually boost user satisfaction. When input is incorporated into the development process, developers can design applications that prioritize their intended users' needs and expectations, enhancing user-centricity. In line with these insights, the proposed approach helps vendors and developers recover from

frequently occurring issues that users confront frequently. However, the approach is still preliminary, we still need to extend it by developing a software prototype that would highlight frequently occurring issues against each identified category, *i.e.*, UI and UX, functionality and features, compatibility and device-specific, customer support and responsiveness, and security and privacy issues group by their occurrence.

## Application of ML and DL in feedback analysis

The utilization of ML and DL algorithms in the examination and classification of user feedback is an essential characteristic that corresponds with prevailing patterns in the field of software engineering research. The mentioned approach holds significant relevance in light of the escalating intricacy and magnitude of user-generated data within digital platforms such as app stores. The implementation of these algorithms, including MNB, LR, RF, and MLP, in the proposed study demonstrates the progress made in the field, as shown in prior research (*Khan et al., 2023a*; *Ullah et al., 2023*). The researchers showcased the application of ML methodologies in the classification and analysis of user comments, thereby making a significant contribution to the advancement of software evolution. The present work expands upon the existing knowledge by incorporating these algorithms and doing a comparative analysis to ascertain their respective effectiveness. This methodological stringency guarantees a more fundamental understanding of how various ML and DL algorithms perform in different techniques of user feedback analysis. In addition, we examine the results of ML and DL models on end-user feedback in the ASA store, which needs to be presented in the software engineering literature. Results show that ML and DL models categorize end-user feedback more accurately.

As a result, they play a vital role in current software analysis as they can process large amounts of user data, which would need to be more effective manually. These algorithms can extract insights from vast and complex datasets (*Ullah et al., 2023*). According to their findings, such models can be used to sift through massive end-user reviews to uncover underlying patterns. This study reflects the sophisticated qualities of current ML and DL performances in the field by preprocessing input data, applying feature engineering, and balancing the data set before implementing these algorithms. The complete process ensures that algorithms process the data while maximizing precision and relevance. This process is essential for ensuring the study's results are relevant and reflect user viewpoints and issues. The use of machine learning and deep learning algorithms in the present study illustrates the changing nature of software engineering research as these technologies become increasingly crucial for analyzing and interpreting user-generated information. Results of the suggested research, which demonstrate sufficient accuracies for MLP, voting, RF, and LSTM classifiers, validate the effectiveness of these refined computational methods for understanding and improving software applications.

## Emphasis on low-rated applications

One of the critical components of the suggested research is examining low-rated apps in the ASA store, which provides an original and vital perspective on software development and user feedback analysis. In contrast to conventional research paradigms, which often

ignore the valuable insight that can be gained from examining low-rated apps in favour of focusing mainly on high-rated apps, this emphasis marks a significant contribution by exploring low-rating applications. As a result of exploring this less-explored research area, the proposed study seeks to shed light on the problems and dissatisfaction associated with software with low ratings. In contrast to the study tendencies noted in *Morales-Ramirez, Kifetew & Perini (2017)* and *Sarro et al. (2018)* which focused primarily on high-rated software programs. The proposed study provides a comprehensive grasp of the factors that lead to consumer dissatisfaction by focusing on low-rated applications and identifying the numerous causes of negative user ratings. Stakeholders and developers can improve software quality and user experience by identifying areas that need improvement. It helps to create a more comprehensive and balanced understanding of user input in app stores by acknowledging that insights from lesser-liked or lower-rated programs are equally valuable for enhancing software products. By bridging a significant gap in knowledge of user preferences and needs in software analysis, this method contributes to the design of more enjoyable software applications.

### Challenges and limitations in utilizing user feedback from low-rated applications

In requirements engineering literature, the importance of end-user feedback in the software evolution process is generally acknowledged by the researchers (*Mao et al., 2017*; *Khan et al., 2019a*; *Ali Khan et al., 2020*). However, very limited research has been reported in the literature on mining feedback information from low-rating software applications. As, high-rating and famous applications are discussed and analyzed more frequently in the literature, while low-rated applications are ignored for possibly gaining some valuable insights for software improvement and evolution (*Ullah et al., 2023*). To date, according to our knowledge, there is no literature reported on low-rating software applications. Still, we identified some insights while analyzing low-rating feedback in the ASA store. For example, the proposed approach classifies end-user feedback into various issue types with considerable performance. However, to make the proposed approach more insightful for software vendors and developers, it must be extended to identify and prioritize actual end-user issues across the various classified issue types. Also, for the low-rating applications in the ASA store, we observed that users submit positive feedback as well, it is challenging to ensure the authenticity of this end-user feedback because looking at the overall rating of the applications and end-user feedback for these apps in which user register severe issues and bugs creates a question on the authenticity of the feedback. Moreover, while analzying the end-user feedback, we recognize that end-users use sarcastic language to report their grudges against the software applications, it becomes challenging to classify sarcastic end-user issues as negative feedback. Therefore, the proposed approach is limited in identifying sarcastic end-user reviews, which we aim to look at in the future to further improve the performance of the proposed approach. It is challenging to counter the end-users bias and attitude towards low-rating applications, as submitting negative feedback might result in a negative experience towards the app on some missing app features or bugs. Another methodological limitation of the proposed approach is that we only considered a small

sample (8,971 reviews) from the ASA store, resulting in identifying six issue categories including UI and UX, functionality and features, compatibility and device-specific, customer support and responsiveness, and security and privacy issues. However, enhancing the review size might result in other issue categories that users report frequently. For this purpose, we are aiming to use unsupervised learning to recover the frequently occurring issue categories and validate the proposed approach by considering a large number of end-user feedback. Additionally, the proposed approach limits in scalability and adaptability, as the ML and DL classifiers show promising results on comparatively a smaller dataset collected from the ASA store, however, its performance still needs to be validated on larger datasets collected from other popular social media platforms, such as app stores, Twitter *etc.*, to identify the generalizability of the proposed approach. Finally, there is a strong need for integrating the proposed approach in the software development and evolution process. It is quite challenging and requires further research on how to integrate the proposed approach into the existing agile software development methodologies to put the valuable insights into practice.

## Addressing core issues in software applications

The research aims to identify and classify the core issues commonly found in software applications with low ratings on ASA stores, which can play a critical role in improving the overall quality and user satisfaction of software apps. The proposed research has focused on the importance of several factors in influencing the user's satisfaction and adoption of software applications, including performance, user UI\UX, functionality, compatibility, customer service, and security-related concerns. The performance and stability of an application are crucial since they directly influence its usefulness and dependability (*Liang et al., 2012*). Users may become dissatisfied and disengaged with the product if these domains are unresolved. An application's user interface UI\UX design is equally crucial, ensuring that it's easy to use, enjoyable, and easily accessible. The software applications should be able to offer the functionality they are provided with quickly and effectively, as users expect. Especially in the current era of technological diversity, ensuring application compatibility across multiple devices and platforms is crucial. When consumers encounter problems or require assistance, providing excellent customer service and adhering to timeliness is essential to maintaining consumer trust and loyalty. Due to the potential for data breaches and privacy infringements to inflict substantial damage upon user confidence and brand standing in the contemporary era of digitalization, security and privacy issues have become increasingly important. By analyzing user comments comprehensively, the research uncovers specific problems and challenges relating to these topics. Therefore, the study can assist developers in prioritizing and addressing these critical domains, ultimately improving software quality and user satisfaction.

Furthermore, the proposed study reveals a preliminary correlation between user interface UI/UX issues, application ratings, and user satisfaction levels based on the manual analysis of the end-user reviews using a content analysis approach. However, it still needs to be elaborated by employing various statistical approaches *i.e.*, Pearson correlation coefficient, spearmans rank correlation coefficient, *etc.* to statistically identify the

correlation between the UI/UX issues, app rating and user satisfaction. Moreover. end-users frequently expressed concerns about the difficulty of navigation, the lack of layout design, the ugly colour schemes, and the lagging interfaces when manually analysing the end-user feedback in the ASA store for various software applications. Also, we are interested in providing further fine-grained analysis to the software developers and vendors by embedding a rule-based analysis to highlight the frequently occurring issues associated with each issue type to give a better visualization of associated issues in the low-rating software applications. It can be concluded based on the critical analysis of the end-user feedback that improving the user interfaces and user experiences is an important component of software development efforts aimed at improving app ratings and user satisfaction. By addressing these identified issues, developers may effectively enhance the overall user experience, increasing user satisfaction and potentially elevating app ratings. UI/UX factors should be considered carefully by software developers during the development process as it is found to be 9.6% of the dataset, representing a commonly occurring issue affecting the quality of software applications. In a competitive digital environment, these elements are crucial to influencing user perceptions and determining the success of their applications.

## Threats to validity

Several potential threats could affect the validity and applicability of the proposed research, and they must be acknowledged and addressed to assess its validity. The proposed study has a limitation, as it is confined to only low-rated applications in the ASA Store and has a limited scope and context. Although insightful, the conclusion may not apply to other digital platforms or applications with higher ratings due to this specific focus. The characteristics of the ASA Store's user base and the nature of the applications it hosts may only represent a subset of the software applications available in other stores. Furthermore, despite their demonstrated efficiency, using advanced ML and DL algorithms for data analysis introduces another source of potential bias. Since these algorithms rely on training data, any inherent biases in the data may skew the results. These ML and DL models may have to incorporate user feedback's intricate and dynamic nature, potentially overlooking subtleties in user sentiment and new trends in software usability and functionality.

In addition, evaluating user input is subjective. Researchers' attitudes and prejudices may influence how reviews are organised and analysed, resulting in variations in results. To ensure that arrangements are as impartial and objective as possible, employing a robust and transparent methodology approach is vital since qualitative research includes subjectivity, such as grounded theory and content analysis. Furthermore, the continually changing environment of software applications and user expectations lends a global dimension to the study's validity. What was true about user input and software difficulties at the time of the study may change or become less relevant as technology and user preferences evolve. Because software development is a dynamic industry, research findings must be updated frequently to guarantee that they remain relevant and valuable in a fast-paced environment. Finally, while the study's methodology is thorough, it may need to be refined to capture the full spectrum of elements influencing user pleasure and application

ratings. Other external factors, such as market trends, competitor actions, and technological advancements that may influence user perceptions and app ratings, must be examined, and our study may need to account for them all. Addressing these validity risks, the study tries to give a fair and full knowledge of the elements influencing the success and failure of low-rated software applications on the ASA Store, while acknowledging the limitations and scope for further research.

In addition, several challenges were encountered during the analysis that required careful consideration to ensure the findings were accurate and robust. The quality of data extracted from ASA Store user reviews presented a significant challenge. Data cleaning and preprocessing techniques were implemented to filter out irrelevant or noisy data, thereby improving the reliability of the analysis. Further, qualitative analysis is subjective, making it challenging to evaluate user input objectively. A rigorous and transparent methodology was used to ensure impartiality and minimize researcher bias in organizing and analyzing user reviews. Despite the inherent challenges encountered during the analysis process, these measures enhanced the accuracy and credibility of the study's findings.

## UI/UX issues limitation

Also, there are some limitations in the proposed study when assessing the validity of the results about UI/UX issues in the software application and their correlation with app ratings and user satisfaction levels. While the proposed approach is an experimental study of identifying frequently occurring issues and offers valuable insights for software vendors and developers, it is still important that UI/UX evaluations are a subjective problem and may be impacted by various variables that the proposed study limits with, including personal preferences and prior experiences. Moreover, the proposed research focuses on textual analysis of user feedback, which could have prevented fully catching the subtleties of feedback linked to UI/UX. Subsequent investigations might use supplementary methodologies like user surveys or usability testing to reinforce the results and tackle these constraints. By using a range of data-gathering techniques, researchers may get a more thorough knowledge of the relationship between UI/UX challenges, app ratings, and user satisfaction levels, enhancing the validity and reliability of their results.

## Ensure reliability and validity

For the proposed research, a series of measures were implemented to ensure the reliability and validity of the classification process for end-user reviews into various issue types. Initially, a fundamental methodological framework was used, using established methodologies such as grounded theory and content analysis to annotate the end-user feedback. These methodologies facilitated the systematic arrangement and analysis of user evaluations, guaranteeing that the conclusions were grounded on empirical evidence and devoid of personal biases. However, the authors who performed the annotation process were also involved in the design and experimental work. Although the annotation process was conducted iteratively and systematically, still there is a chance that the coders have unintentionally attempted to make a second guess. In addition, to ensure the proposed process's validity and enhance the trustworthiness of the findings, a negotiation process

and implementation of inter-rater reliability checks were in place. Moreover, measures were taken to guarantee openness and uniformity throughout the analytical procedure to mitigate the inherent subjectivity of user perspectives. The study included thorough documentation of the criteria used for categorization and the methods used for decision-making. Additionally, frequent peer reviews were conducted to verify the results' validity. In addition, rigorous training and validation protocols were implemented to mitigate any biases introduced by sophisticated ML and DL algorithms, and the algorithms' efficacy and precision were consistently evaluated using various evaluation matrices, including, accuracy, precision, recall, f-measure, RUC, AUC, Confusion matrix, and training validation loss.

## Generalizability of the findings

The proposed research's primary objective is to examine end-user feedback in the ASA store to classify end-user feedback into frequently occurring issue types. The results demonstrate its better performance on the dataset collected from the ASA store. However, as previously mentioned the proposed approach is limited in scalability and adaptability, as the ML and DL classifiers show promising results on comparatively a smaller dataset collected from the ASA store, its performance still needs to be validated on larger datasets collected from other popular social media platforms, such as app stores, Twitter *etc.*, to identify the generalizability of the proposed approach. Therefore, it is critical to consider the applicability of the proposed findings to various mobile apps and app stores such as Google and Apple Play Store. To address this issue, we need to curate a dataset comprising end-user feedback from multiple social media platforms. In addition, we believe that the proposed fine-tuned approach can be applied to the end-user feedback collected from the iPhone and Google Play store to identify its generality. In the proposed approach, we thoroughly investigated the end-user feedback to identify issues categories that end-user reports in the ASA store. Therefore, we are confident in the assumption that the proposed approach can be generalised in classifying end-user feedback collected from app stores into various issue types. Additionally, we explore the broader implications of the proposed study, including prevailing trends in user feedback and the significance of specific problem categories detected in the ASA store for other platforms. The insights derived from our investigation indicate that numerous mobile apps exhibit recurring problem categories, such as performance, UX/UI, functionality, compatibility, customer service, and security concerns. The findings of this study suggest that although our research specifically examined a specific app store, the challenges and concerns users voice are likely to reflect broader trends in the mobile app platforms. Performance, usability, functionality, and security are pivotal in influencing user satisfaction and involvement beyond the boundaries of particular application markets. Therefore, while our analysis primarily focuses on the ASA store, the insights obtained can provide valuable information to developers and stakeholders working on many platforms. To enhance the overall quality and user experience of mobile apps, developers should recognize the common trends in user feedback dynamics and prioritize the identified areas of concern independent of the app store on which they are housed.

## CONCLUSION AND FUTURE WORK

In this research study, we propose a comprehensive study of low-rated software applications in the ASA store, highlighting the critical role of end-user feedback in software evolution and continuous improvement. The study fills a substantial gap by focusing on underexplored but equally important low-rated software applications from less focused ASA stores, resulting in a better understanding of user dissatisfaction and the variables that contribute to lower ratings. This approach differs from the existing literature's emphasis on high-rated software applications. Grounded theory and content analysis have been useful in classifying and analyzing user feedback, depending on well-established approaches in the field. As utilized by researchers like *Strauss & Corbin (1990)* and *Neuendorf (2017)*, this approach allows for a frequent examination of end-user reviews into actionable issue categories. The insights gained from this research spanning performance, UI/UX, functionality, compatibility, customer support, and security issues are instrumental in pinpointing areas for improvement. The use of ML and DL algorithms, such as MNB, LR, RF, MLP, KNN, AdaBoost, Voting, CNN, GRU, BiGRU, LSTM, BILSTM, and RNN classifiers, shows the promise of these technologies for processing and analyzing big datasets. The remarkable accuracy rates created by these algorithms show their capacity to derive important patterns from end-user feedback. This agrees with the existing literature (*Ullah et al., 2023*; *Khan et al., 2022*, *2023a*), which has likewise used ML approaches for user feedback analysis. The significance of this finding is significant. It delivers a roadmap for developers and requirements engineers to fix the inadequacies in low-rated apps, resulting in improved user satisfaction and perhaps enhanced ratings by including the identified commonly occurring issues in the software evolution process. It delivers academics with a new approach to evaluating user reviews, promoting a change in focus from highly rated-to low-ranked applications. This study consequently contributes to the larger conversation about software quality and user-friendly development in the digital age. This study adds to the existing understanding of app store statistics and introduces new options for further investigation. It is possible for further studies to delve into the longitudinal effect of gains made based on user feedback and to explore the interplay between user ratings and the evolving nature of software applications. In conclusion, this study emphasizes the underutilized potential of low-rated software applications as a source of useful insights for software enhancement. Using a variety of grounded theories, content analysis, and advanced computational approaches, provides a strong foundation for understanding and addressing the issues these applications experience, eventually contributing to improving software quality and the user experience in the app stores.

Furthermore, it is crucial to consider the applicability of our outcomes outside the ASA Store and the particular issue areas represented in our research. The current study significantly contributes to understanding user feedback patterns and prevalent problems inside the ASA Store. However, it is recommended that future research efforts focus on verifying and developing these findings across various mobile applications and app stores. Examining multiple platforms and places through proximate analysis can provide insights

into the parallels and differences in user expectations and satisfaction levels. This, in turn, contributes to a more complete comprehension of the dynamics of software quality in the broader app ecosystem. Researchers can improve the applicability of our findings to a more comprehensive range of software applications and platforms by analyzing user feedback using many techniques to specify overarching themes and the best approaches for managing user issues.

This study's investigation of low-rated software applications in the ASA store, primarily through end-user feedback analysis and the implementation of ML and DL algorithms, opens several ways for future research. The first area of future investigation is to have a better understanding of end-user input. Current research, including our own, is mostly concerned with textual analysis of user reviews of low-ranked software products on the ASA store. Future research could combine sentiment analysis and emotional intelligence into ML and DL models to improve the performance of nuanced sentiments represented in user evaluations. Integrating models that can understand sarcasm, humor, and context-specific language nuances may improve data processing. We may look at constructing more complex Natural Language Processing (NLP) models that can detect slight frictions in user sentiments. This improvement would provide a more detailed insight into customer displeasure or specific application problem areas. Another promising direction is extending this research to a broader range of applications and platforms. While our study focused on the ASA Store, future research might include applications from other platforms, such as Google Play and Apple's App Store, to evaluate user opinion across Play Stores. This broader approach may mine platform-specific faults or deliver insights into how different user groups perceive app quality. Similarly, including other geographical regions and languages in the research may provide a more global perspective on app performance and user happiness. The connection between app updates, user ratings, and feedback changes is also worth investigating. Understanding how individual upgrades (feature additions, UI changes, and bug patches) affect user satisfaction may offer developers actionable information for future releases. ML and DL algorithms could be used to predict user reactions to upcoming updates or to recommend the most significant areas for development based on past data.

### Funding
The authors received no funding for this work.

### Competing Interests
The authors declare that they have no competing interests.

### Author Contributions
- Nek Dil Khan conceived and designed the experiments, performed the experiments, analyzed the data, performed the computation work, authored or reviewed drafts of the article, and approved the final draft.

• Javed Ali Khan conceived and designed the experiments, performed the experiments, analyzed the data, performed the computation work, authored or reviewed drafts of the article, and approved the final draft.
• Jianqiang Li analyzed the data, authored or reviewed drafts of the article, and approved the final draft.
• Tahir Ullah performed the experiments, analyzed the data, prepared figures and/or tables, authored or reviewed drafts of the article, and approved the final draft.
• Qing Zhao performed the experiments, analyzed the data, prepared figures and/or tables, authored or reviewed drafts of the article, and approved the final draft.

## Data Availability

The data and models are available at GitHub and Zenodo:

- https://github.com/nekdil566/issue-detection.

- Beijing University of Technology, & Khan, N. D. (2024). Mining software insights: uncovering the frequently occurring issues in low-rating software applications. https://doi.org/10.5281/zenodo.11256608.

## Supplemental Information

Supplemental information for this article can be found online at http://dx.doi.org/10.7717/peerj-cs.2115#supplemental-information.

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
