# Peer review of "Mining software insights: uncovering the frequently occurring issues in low-rating software applications"

_PeerJ Computer Science, doi:10.7717/peerj-cs.2115_

## Round 0.1 · original submission · Major Revisions

I have received the review reports for your paper submitted to PeerJ Computer Science from the reviewers. According to the reports, I will recommend major revision to your paper. Please refer to the reviewers’ opinions to improve your paper. Please also write a revision note such that the reviewers can easily check whether their comments are fully addressed. We look forward to receiving your revised manuscript soon.

Reviewer 1 ·

Basic reporting

This study highlights the importance of user feedback in software development, particularly in addressing negative reviews in App stores. Methodologically, it employs grounded theory and content analysis, coupled with machine learning algorithms, to analyze and improve low-rated applications based on end-user feedback. The paper is good but requires revision and comments are provided.

The abstract is too long. Please re-write an abstract section, explain an obtained result and contribution, improve a proposed method, etc. Please delete unnecessary information.

The novelty of this paper needs to be revised. How do the findings from this study align with or differ from previous research regarding the performance of ML and DL classifiers in multi-class classification tasks, particularly in the context of software reviews?

Discuss the prevailing challenges and limitations in the existing literature regarding the utilization of user feedback from low-rated applications for software improvement.

Experimental design

Please elaborate more on measures taken to ensure the reliability and validity of the categorization process for end-user feedback, considering the subjective nature of user perceptions.

Do UI/UX issues identified in the end-user feedback correlate with app ratings and user satisfaction levels, and what specific design elements are frequently criticized by users?

Can you add more insights that can be gained from the comparison of performance metrics across different ML and DL classifiers, particularly regarding their effectiveness in classifying specific issue types such as compatibility, functionality, customer support, security, and UI/UX issues?

Validity of the findings

Please discuss to what extent can the findings from this study be generalized to other mobile applications and app stores, considering the focus on the ASA store and specific issue categories.

It is not very clear how the collected end-user reviews contribute to a more nuanced understanding of user preferences, issues, and overall experiences with software applications, and what general trends or patterns emerge from the dataset.

What were the specific improvements observed in the performance of DL classifiers, such as CNN, GRU, BiGRU, LSTM, BILSTM, and RNN, upon adding dropout layers with a dropout rate of 0.2, and how did this enhance their ability to extract issue types while reducing overfitting?

How did the utilization of different DL optimizers, such as Adam and RMSProp, impact the model accuracy in classifying various issue types, and what insights can be gleaned from comparing their performance outcomes?

Additional comments

What challenges or limitations were encountered during the analysis process, and how were they addressed to ensure the accuracy and robustness of the findings?

Reviewer 2 ·

Basic reporting

The article includes sufficient introduction and background to demonstrate how the work fits into the broader field of knowledge.

Experimental design

Methods are described with sufficient information to be reproducible by another investigator.

Validity of the findings

no comment

Additional comments

This manuscript introduces a compelling approach to analyzing end-user evaluations, specifically focusing on lower ratings, and categorizes them into distinct issues. This methodology aids developers in enhancing both the application’s functionality and user satisfaction. The manuscript is clearly articulated and structured for easy comprehension. Nonetheless, there are several aspects that warrant attention:

1) The sequence of citations requires rectification to ensure they are in ascending order; for instance, the current sequence [17] [18] [19] [20] [12] should be revised to [12] [17] [18] [19] [20].
2) The paper posits three research questions, yet it remains challenging to delineate which experiments substantiate each question.
3) Throughout the experimental analysis, it is observed that models such as MLP, RF, and BiGRU, GRU consistently outperform BiLSTM and LSTM. Given that MLP and RF are less complex than BiLSTM and LSTM, and considering that BiGRU and GRU are simplified variants of BiLSTM and LSTM, it raises the question of whether the dataset is too limited for effective deep learning application. It is advisable for the authors to delve deeper into the performance analysis of the algorithms as presented in Table 5, to provide more nuanced insights into their effectiveness.

---

## Round 0.2 · accepted · Accept

The authors have addressed the comments of the reviewers. I am happy to make a decision of acceptance to the paper.

Reviewer 2 ·

Basic reporting

Literature references, sufficient field background/context provided.

Experimental design

Research question well defined, relevant & meaningful. It is stated how research fills an identified knowledge gap.

Validity of the findings

All underlying data have been provided; they are robust, statistically sound, & controlled.

Additional comments

The authors have explained all reviewers'feedback. I think this version is good enough.